

# Error propagation in spectrometric functions of soil organic carbon

Monja Ellinger [a], Ines Merbach [b], Ulrike Werban[c], Mareike Ließ [a]

[a] Department Soil System Science, Helmholtz Centre for Environmental Research – UFZ, Halle (Saale), Germany

[b] Department Community Ecology, Helmholtz Centre for Environmental Research – UFZ, Bad Lauchstädt, Germany

[c] Department Monitoring & Exploration Technologies, Helmholtz Centre for Environmental Research – UFZ, Leipzig, Germany

*Correspondence to:* Mareike Ließ (mareike.liess@ufz.de)

**Abstract**

Soil organic carbon (SOC) plays a major role concerning the chemical, physical and biological soil properties and functions. To get a better understanding how soil management affects the SOC content, the exact monitoring of SOC on long-term field experiments (LTFE) is needed. Visible and near infrared (Vis-NIR) reflectance spectrometry is an inexpensive and fast possibility to enhance conventional SOC analysis and has often been used to predict SOC. For this study, 100 soil samples were collected at a LTFE in central Germany by two different sampling designs. Regression models were built using partial least square regression (PLSR). In order to build robust models, 10-fold cross-validation was used for model tuning and validation procedure. We analysed and discussed various aspects that influence the obtained error measure. A transparent and precise documentation of the model building and validation process, including the calculation of the error measure, is necessary in order to assess the model accuracy in a comprehensive way. This would be the first step to gain a standardized method for model building and validation procedure.

Keywords: Soil organic carbon, Laboratory Vis-NIR reflectance spectrometry, Partial least square regression, repeated 10-fold cross-validation



## 1    Introduction

Soil is at the same time one of the most important and one of the most limited natural resources. Most of all it is

needed for food production, but also for the production of energy and fibre or for the provision of fresh water

(Johnson, 2008; Lorenz and Lal, 2016; Stenberg et al., 2010). All these functions depend on the quality of the

existing soil. This quality in turn is much influenced by its SOC content since it affects chemical, physical and

biological soil properties and functions (Knadel et al., 2015; Lorenz and Lal, 2016). Additionally, SOC is also

interesting when it comes to the global warming issue since soil is the largest terrestrial reservoir of organic

carbon in the world (Conforti et al., 2015; Johnson, 2008; McBratney et al., 2014; Stockmann et al., 2013). The

SOC content of soils can be increased through the sequestration of atmospheric $CO_2$ into long-living components

of soils (Lal, 2004; McBratney et al., 2014). Thus, the SOC stock of soils could be used as a manageable sink for

atmospheric carbon (Stockmann et al., 2013), achieving both food security and a strategy against the increasing

$CO_2$-concentration in the global atmosphere (Lal, 2004; Lorenz and Lal, 2016; McBratney et al., 2014). As the

SOC content of soils reacts very slowly to environmental changes (Meersmans et al., 2009), long-term field

experiments (LTFE) are required to understand the impact of soil management and farming systems on the rate

of SOC sequestration (Lal, 2004) as well as on yield and crop quality in the long run. The precise monitoring of

SOC on a LTFE with conventional lab analysis is labour-intensive and expensive (Adamchuk and Viscarra

Rossel, 2010; Loum et al., 2016) as it requires the analysis of a rather high amount of samples. Visible and near

infrared (Vis-NIR) reflectance spectrometry can facilitate this procedure. It is non-destructive, fast and

economical (Mouazen et al., 2010; Tekin et al., 2014), requiring only a small number of soil samples and little

sample preparation (Conforti et al., 2015). In addition, no chemicals are needed and one spectrum contains

information about many different soil components (Conforti et al., 2015; Viscarra Rossel et al., 2006b). Vis-NIR

soil spectrometry has been used on many occasions to build SOC prediction models (Jiang et al., 2016; Kuang

and Mouazen, 2013; Nocita et al., 2013). However within every model building process, uncertainties of the

input data affect the overall model uncertainty (error propagation). Those uncertainties contain measurement

errors as well as variation in the input data (Heuvelink, 1999). In most studies dealing with SOC prediction from

Vis-NIR spectra, no clear statement about these uncertainties or their handling is made. Up to now the general

approach consisted of averaging the multiple measured spectra of one sample to one spectrum which was then

used for model building (Ge et al., 2011; Stevens et al., 2013; Viscarra Rossel et al., 2003). But the number of

measurements used to gain one averaged spectrum differs between studies. Jiang et al. (2016), for example,

averaged 10 measurements to receive one spectrum, while Volkan Bilgili et al. (2010) and Wang et al. (2014)

used four measurements. This difference is assumed to have an influence on the uncertainties implemented in the



input data. Besides input data uncertainties there are many other aspects influencing the model building process

such as the sampling design and the chosen pre-processing method. This study aims to investigate the impact of

these aspects on the model outcome and the calculation of error measures. Also, the influence of the model

tuning procedure and the data sub setting and / or resampling for the validation procedure will be discussed.

## 2  Material and Methods

### 2.1  The static fertilization experiment Bad Lauchstädt

The soil samples were taken at the LTFE site "Static Fertilisation Experiment" in Bad Lauchstädt in central

German (Körschens and Pfefferkorn, 1998). Positioned at 51° 24' N, 11 ° 53' E and with an altitude of 113m

(Körschens and Pfefferkorn, 1998), the climate is characterized by an mean annual precipitation of 470 – 540

mm and an average mean annual temperature of 8.5 – 9.0 °C. The soil type was characterized as a haplic

Chernozem developed from loess (Altermann et al., 2005) with a soil texture of 21.1 ± 1.2 % clay, 72.1 ± 1.7 %

silt, and 6.9 ± 1.9 % sand (Dierke and Werban, 2013). Saturated water conductivity and air capacity are medium

to high in the top soil (Altermann et al., 2005). The Static Fertilization Experiment was initialized in 1902 by

Schneidewind and Gröbler and is about 4 ha in size (Merbach and Schulz, 2013) Its objective is to investigate

the impact of organic and mineral fertilization on soil fertility as well as yield and quality of crops (Körschens

and Pfefferkorn, 1998; Schulz, 2017). The experiment includes eight subfields with a width from 25.2 m to 28.5

m and a length of 190 m which are each divided into 18 plots that are treated with different mineral and organic

fertilizer as well as planted with different crops following a crop rotation (Körschens and Pfefferkorn, 1998).

The plots of subfields 4 and 5 are additionally parted into 5 smaller subplots.

### 2.2  Sampling design

A total of 100 soil samples were taken at the soil surface (0-10 cm) in September 2016. The exact location of the

sampling points was determined by a differential GPS GNSS LEICA Viva GS08. It was decided to sample at

precise point locations instead of taking samples representative for LTFE plots to allow for a direct comparison

with spectrometric field measurements for area-wide regionalisation (not included in this study). The sampling

points were determined beforehand by two sampling designs. Based on the LTFE treatment factors and per-plot

soil archive data including $C_{org}$, $N_{tot}$, plant available P, plant available K (both with DL-Method (VDLUFA,

2012))  and pH (Fig. 1) both designs strived to select a dataset of 50 samples representative for the soil

variability of the entire LTFE. Categorical and continuous data first entered a factor analysis with mixed data

(FAMD) performed with R package FactoMineR (Lê et al., 2008) to allow for further joint analysis. For design



'A' the LTFE plots were then grouped by a k-means cluster analysis. R package NbClust (Charrad et al., 2014) automatically determines the optimal number of clusters making use of 30 indices. In the end, ten plots were

randomly selected from each of the resulting five clusters making a total of 50 plots to be sampled. For design 'B' the Kennard-Stone algorithm was applied with R package prospectr (Kennard and Stone, 1969; Stevens and Ramirez Lopez, 2014). 50 LTFE plots were selected involving 5 repetitions of the algorithm to reduce inter-point dependence. Finally, one sampling point was randomly selected from each of the 50 LTFE plots from design A and B based on a 5 x 5 cm raster. Plot margins of 1.5 m (3 m between plots) were excluded. Fig. 2

shows the location of the so obtained 100 soil samples.

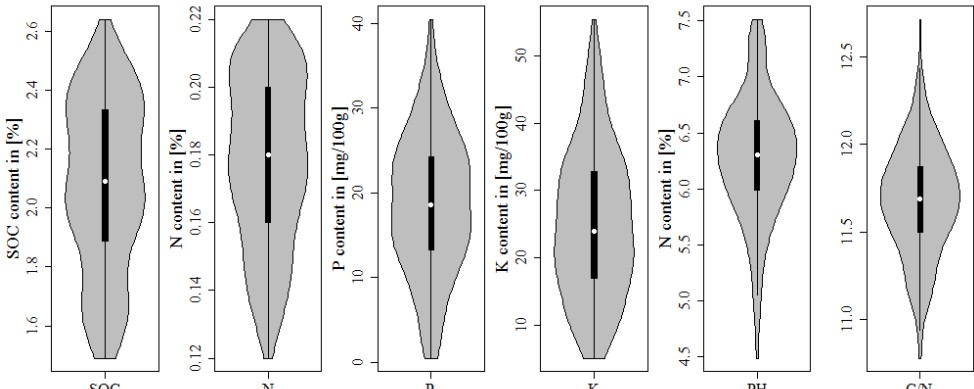

**Fig. 1 Soil archive data of the LTFE measured from 2004 to 2007 (Reports of the experimental station Bad Lauschstädt 2004-2007 (unpublished)).**





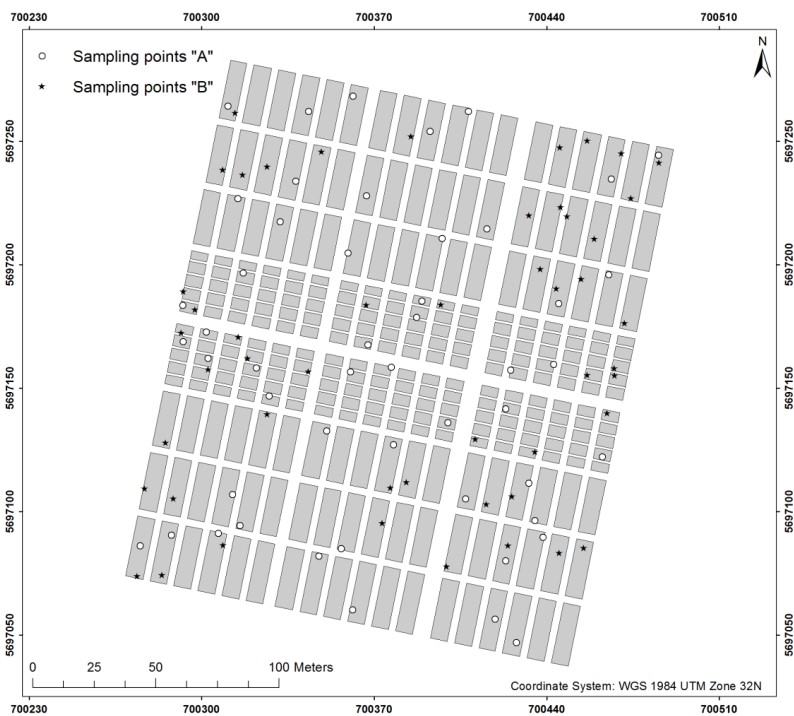

**Fig. 2 Site of the static Fertilisation Experiment in Bad Lauchstädt with LTFE plots and sampling points according to design A and B. Plot margins excluded from sampling are visible as 3 m wide stripes between plots.**

### 2.3 Laboratory measurements

The soil samples were air-dried, sieved and grinded prior to C/N - measurements with dry combustion. Measurements were repeated in three replicate samples. C measurements were taken as organic carbon due to negligibly small carbonate contents. The Vis-NIR contact measurements were performed on air-dried and sieved (2 mm) samples in July 2017, using Veris® VIS-NIR Spectrophotometer by Veris technologies, Inc. (hereinafter called Veris) containing a Hamamatsu Mini-spectrometer TG series (900 to 2550 nm) and an Ocean Optics USB4000 instrument (200 to 1100 nm). The device was warmed up for at least 20 minutes before performing measurements. All measurements were taken in a dark room to prevent daylight from affecting the outcome. The soil samples were scanned from the top. Before and between soil sample measurements, Veris was calibrated using four external references. Each soil sample was divided into three sub-samples filled into petri dishes (Schott Duran petri dishes; Duran Group, Mainz, Germany). These sub-samples were not related to the three replicate samples used for C/N-measurements. For each sub-sample six spectra were gained by measuring it





three times, rotating it by 90 degrees and then measuring it three times again. This procedure resulted in 18

spectra for each soil sample.

## 2.4 Spectral pre-processing

Veris is equipped with two spectrometers. At the beginning and end of their respective wavelength ranges noise

occurs in the measurements. Therefore, the spectra between these wavelengths (1000 to 1100 nm) had to be

removed. Additionally, the spectra were cut at the beginning (402 nm) and the end (2220 nm) to remove noise. A

number of pre-processing methods were tested to enhance the information regarding SOC in the Vis-NIR

spectra. Spectral absorption features are caused by vibrational stretching and bending of structural molecule

groups and electronic excitation (Ben-Dor et al., 1999; Dalal and Henry, 1986). Molecule vibrations from

hydroxyl, carboxyl and amine functional groups produce soil absorption features related to soil organic matter in

the mid-infrared (MIR) region of the spectra (Croft et al., 2012).Vis-NIR spectra show only broad and unclear

adsorption features related to overtone vibrations from the MIR (Stenberg and Viscarra Rossel, 2010; Viscarra

Rossel et al., 2006a). For this reason, the application of pre-processing methods to Vis-NIR spectra is necessary

in order to gain soil property related information (Stenberg and Viscarra Rossel, 2010). The spectra were tested

for outliers using R package mvoutlier (Filzmoser and Gschwandtner, 2017). For this procedure a PCA is

performed, using then the first two obtained PCs for outlier detection with function aq.plot. According to

Filzmoser (2005), the Mahalanobis distance of normally distributed data follows a chi-square distribution.

Observations which lay beyond a certain quantile of this distribution are marked as outliers and removed from

the data (Filzmoser and Gschwandtner, 2017). In this study, no outliers were detected. Out of the tested pre-

processing methods, four different combinations are shown in this study in order to demonstrate their different

effects on the prediction model. Their application resulted  in spectra with different wavelength ranges (Table 1)

and different appearance (Fig. 3). These pre-processing techniques include the Savitzky-Golay algorithm (SG),

continuum removal (CR), the standard normal variate (SNV), the first derivative (d1) and the gap-segment

algorithm (gapDer). The SG algorithm fits a polynomial regression on the spectral data to find the derivative at a

center point i of a defined smoothing window (w) (Rinnan et al., 2009; Savitzky and Golay, 1964; Swarbrick,

2016). CR can be seen as a spectra normalization technique which enables to compare different absorption

characteristics from a mutual baseline (Kokaly, 2001; Mutanga and Skidmore, 2003). It identifies the local

reflectance spectra maximum points and connects those points to form a convex hull (Mutanga and Skidmore,

2003; Stevens and Ramirez Lopez, 2014). Calculating



$$\phi_i = \frac{x_i}{c_i} \qquad (1)$$

for i = {1, … , p) with $x_i$ and $c_i$ being the initial and the continuum reflectance values at wavelength i of a set of p
wavelengths then gives the continuum-removed reflectance value $\phi i$ (Stevens and Ramirez Lopez, 2014). All

other data have values between 1 and 0 (Mutanga and Skidmore, 2003; Schmidt and Skidmore, 2001). Thus the

absorption peaks are enhanced (Schmidt and Skidmore, 2001). SNV is a scatter-corrective pre-processing

method (Rinnan et al., 2009).

The basic formula is as follows

$$x_{corr} = \frac{x_{org} - a_0}{a_1} \qquad (2)$$

with $a_0$ as the measured spectrum's average value which shall be corrected and $a_1$ being the sample spectrum's

standard deviation. $x_{org}$ are the original spectra and $x_{corr}$ the corrected spectra after applying SNV. SNV operates

row-wise, so each observation is processed on its own (Rinnan et al., 2009; Stevens and Ramirez Lopez, 2014).

d1 represents the slope of the spectrum, showing peaks where the spectrum displays its maximum slope and

crossing zero where the spectrum shows peaks (Leone et al., 2012). According to Knadel et al. (2015) and  Smith

(2002), d1 can be used to remove baseline offsets from the spectra. The estimation of d1 is done by computing

the difference between two batched spectral points $x_i$ and $x_{i-1}$ (Eq. 3)

$$x_i{}' = x_i - x_{i-1} \qquad (3)$$

with $x_i{}'$ as the value of the first derivative at the $i^{th}$ wavelength (Rinnan et al., 2009). The downside of using

derivative spectra is their tendency to over-fit the calibration model (Stevens and Ramirez Lopez, 2014).

Moreover, derivatives may increase noise so that a smoothing of the data is required (Leone et al., 2012; Stevens

and Ramirez Lopez, 2014). With the gapDer a smoothing is performed under a chosen segment size (s) and then

a derivative follows (Stevens and Ramirez Lopez, 2014). The application of the different pre-processing methods

for this study was done using R package prospectr (Stevens and Ramirez Lopez, 2014).

**Table 1 Combinations of pre-processing techniques used in this study; w = window size, s = segment size.**

| Pre-processing methods | Wavelength range | Abbr. |
| --- | --- | --- |
| Savitzky-Golay (w =11 nm) | 432 – 2201 nm | SG |



| Savitzky-Golay (w=11 nm) and continuum removal | 432 – 2201 nm | SGCR |
| Standard normal variate and 1$^{st}$ derivative | 408 – 2186 nm | SNVd1 |
| Gap-segment algorithm (w = 11 nm, s = 10 nm) | 490 – 2163 nm | gapDer |



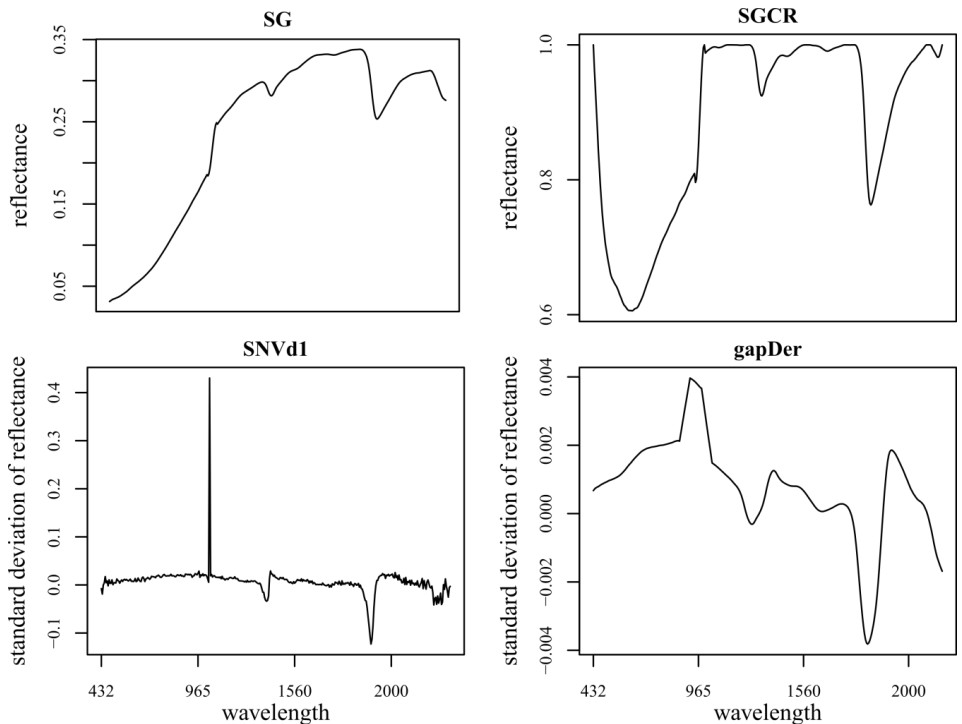

**Fig. 3 Impact of different pre-processing techniques on a spectrum; SG = Savitzky-Golay, CR = continuum removal,
SNV = standard normal variate, d1 = 1$^{st}$ derivative, gapDer = Gap-segment algorithm.**

## 2.5 Error propagation

A problem occurring in every model building process is uncertainty propagation. Where uncertainties already
existing in the input data as well as in the used model result in uncertainties in the output (Brown and Heuvelink,
2006). Uncertainties in the input data are caused by errors in data acquisition (e.g. measurement errors) as well
as variation in the data themselves (e.g. within-sample variability) (Heuvelink, 1999). For this study, there are



two different sources for errors in data acquisition: the measurement of the spectral data and the measurement of

the SOC content of the soil samples. In order to investigate the influence of these errors, different data sets were

built in this study. Fig. 4 gives an overview. From the measured Vis-NIR spectra, three different spectral data

variants were created (Fig. 4, step 1). For the first variant, all 18 spectra were retained. The inclusion of all 18

spectra reveals the influence of the error implemented in the spectral measurements as well as the influence of

the within-sample variability. For the second, the three measurements obtained before and after sample rotation

were averaged separately resulting in 6 spectra per sample showing the influence of within-sample variability.

For the third data variant, all 18 spectra were averaged to 1 mean spectrum per sample, removing the influence

of the measurement error as well as the within-sample variability. The different spectra obtained through this

procedure can be seen in Fig. 5. Only parts of the spectra are depicted in order to show their differences. The

three different spectral data variants were then pre-processed with the different pre-processing methods from

Table 1 (Fig. 4, step 2), resulting into 12 different spectral data sets (Fig. 4, step 3). These were then combined

with single and averaged SOC values in step 4 so that altogether 24 data sets were obtained (Fig. 4, step 5). In

order to compare the two sampling designs, this procedure was carried out for the 50 soil samples labelled "A"

and "B" and also for the complete set of soil samples. In this way, three different soil sample sets ("A", "B" and

"all samples") were achieved.

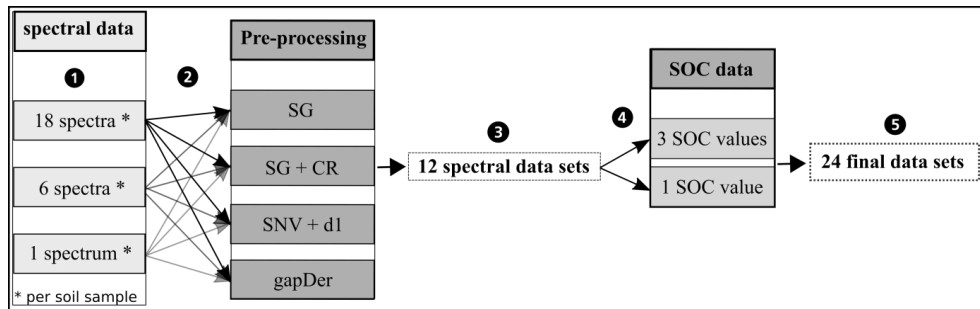


**Fig. 4 Data sets to investigate the uncertainty propagation.**



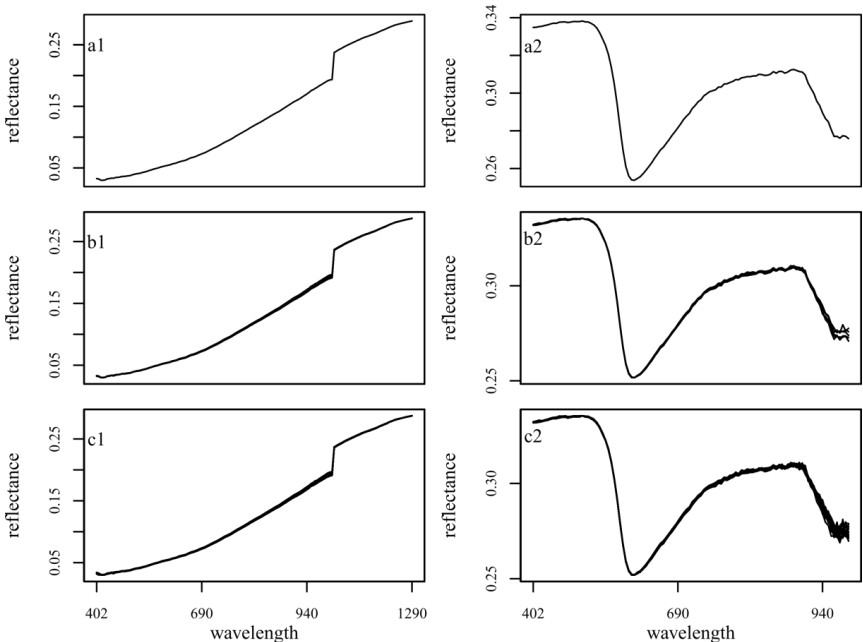

**Fig. 5 Different sets of spectral data per soil sample: a1 and a2 showing 1 mean spectrum, b1 und b2 6 averaged spectra and c1 and c2 18 original spectra.**

### 2.6    Model building and validation

Regression models were built using partial least square regression (PLSR). Out of the many algorithms, PLSR is seen as a standard method for spectral calibration and prediction (Mouazen et al., 2010; Tekin et al., 2014; Viscarra Rossel et al., 2006b). For applications to predict SOC from Vis-NIR soil spectra see (Conforti et al., 2015; Jiang et al., 2016; Kuang and Mouazen, 2013; Nocita et al., 2013). PLSR is described in detail by Martens and Næs (1989) and Naes et al. (2002). It incorporates characteristics from principle component analysis (PCA) and multiple regression (Abdi, 2007). The concept behind PLSR is to seek a small number of linear combinations (components or latent factors) obtained from the measured spectral data and to use them in the regression equation to predict SOC instead of the initial values (Martens and Næs, 1989; Naes et al., 2002). These components are constructed so that they account for most variance in the measured spectral data (X) and the SOC content (Y) and at the same time maximize the correlation between X and Y. In other words, PLSR leads to the covariance between X and Y being maximized (Bjørsvik and Martens, 2008; Summers et al., 2011; Tekin et al., 2014; Wehrens, 2011).

In order to receive a robust model, it is important not to include too many components in model building as this will lead to over-fitting (Hastie et al., 2009; Kuhn and Johnson, 2013). On the other hand, the inclusion of too



few components comprises the risk of building an under-fitted model which is too small to cover the variability

existing in the soil spectral data (Naes et al., 2002). The selection of the optimal number of components is

hereinafter referred to as *model tuning*. In order to receive a robust model, resampling is commonly applied for

model building and validation. But resampling can also be used for *model tuning* to receive robust tuning

parameters (Guio Blanco et al., 2018; Hastie et al., 2009; Kuhn and Johnson, 2013). For small data sets, k-fold

cross-validation is recommended (Hastie et al., 2009).

In this study, model building, *model validation* and *model tuning* was done using repeated 10-fold cross-

validation (10-fold CV), an approach applied before by (Guio Blanco et al., 2018). Repeated 10-fold CV can

increase the precision of the prediction while maintaining a small bias (Kuhn and Johnson, 2013). Five

Repetitions of 10-fold CV were conducted in this case.

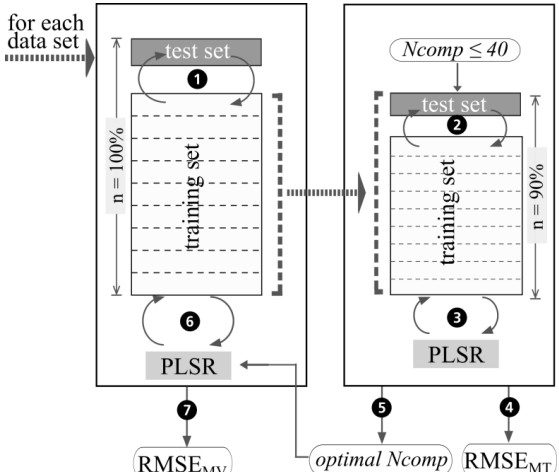


**Fig. 6 M*odel tuning* and *model validation* procedure. The right box shows the *model tuning*, the left one the *model validation* procedure; Ncomp = number of components; adapted from Guio Blanco et al. (2018).**

Fig. 6 shows the various steps of the modelling procedure involving repeated 10-fold CV for *model tuning* (right

box) and *validation* (left box). In the process the data set (n = 100%) is divided into 10 folds of equal size (step

1). This was done using the sample() function in R from package dplyr (Wickham and Francois, 2017). One of

the 10 folds is held out as test set and the other nine are used as training set and partitioned again into 10 folds

for *model tuning* (step 2). The optimal number of components (*optimal Ncomp*) is then determined by computing

a PLSR on the resampled data testing 1 to 40 components (step 3) calculating the repeatedly 10-fold cross-

validated RMSE of *model tuning* ($RMSE_{MT}$) (step 4). This was implemented with the trainControl() function in

package caret (Kuhn, 2017). The optimal number of components (step 5) is then used in model building for

validation (step 6) with PLSR using train() function from caret. The resulting model's test set RMSE of model



validation ($RMSE_{MV}$) is determined in step 7. The whole procedure is repeated until all folds in the boxes had once been used as the test set.

In this study, the partition of the spectral data into 10 folds for 10-fold CV had to be done very carefully, as in
some cases multiple spectra existed for one sample. In order to reduce autocorrelation, it was especially important to ensure that those spectra and also spectra gained from samples which were located in the same treatment plot were always in the same fold during 10-fold CV, as those spectra were assumed to be very similar or even identical. Thus, it was possible to compare the results gained with the different data sets and to investigate the influence of uncertainties implemented in the input data.

**3       Results and Discussion**

**3.1      Soil organic carbon content**

Fig. 7 shows the distribution of the SOC content of the three soil sample sets, consisting of 50 soil samples labelled "A" and "B" and  100 soil samples referred to as "all samples". The measurement error existing in the SOC measurements, here defined as the difference between replicate measurements, ranges from 0.003 to 0.229
% SOC with a mean of 0.048 % SOC. The aim of this study was not to compare the two different sampling designs among each other, but to test whether they are representative of the SOC values existing on the LTFE. For this purpose, per-plot soil archive data from the years 2004 to 2007 are also displayed. The statistics of the data are given in Table 2. In order to compare distributions between the archive SOC data and "A", "B" and "all samples", a Mann–Whitney U test was applied to the data, testing the "A", "B" and "all samples" against the
archive data, respectively. In all cases, no significant difference between the different sample sets and the archive data could be found. This shows that all soil sample sets used in this study are representative for the SOC values existing in the LTFE. Nevertheless, the SOC distribution of "A" and "B" samples differ, with the "A" samples resembling the distribution of all 100 samples more than the "B" samples. The "A" samples contained more samples representing higher SOC values, whereas the "B" samples show a higher representation for lower SOC
values. This difference in the distribution of SOC values may have an influence on the prediction results of the models built with "A" and "B" samples. "A" models may be better in predicting higher SOC values, while simultaneously failing to estimate lower SOC values in an appropriate way. To the contrary, "B" models may predict lower SOC values more accurate than higher SOC values. The violin plots of all three soil sample sets do not resembles the archive violin plot very much. The plots for "A" and all samples show higher and lower SOC
values than the archive data due to the fact that those data are obtained from compound samples for one plot. The



"B" samples share the same minimum value with the archive data, but display slightly higher SOC values. This shows that the choice of the sampling design has an influence on the model outcome, even if both designs represent the SOC values on the experimental field in an appropriate way.

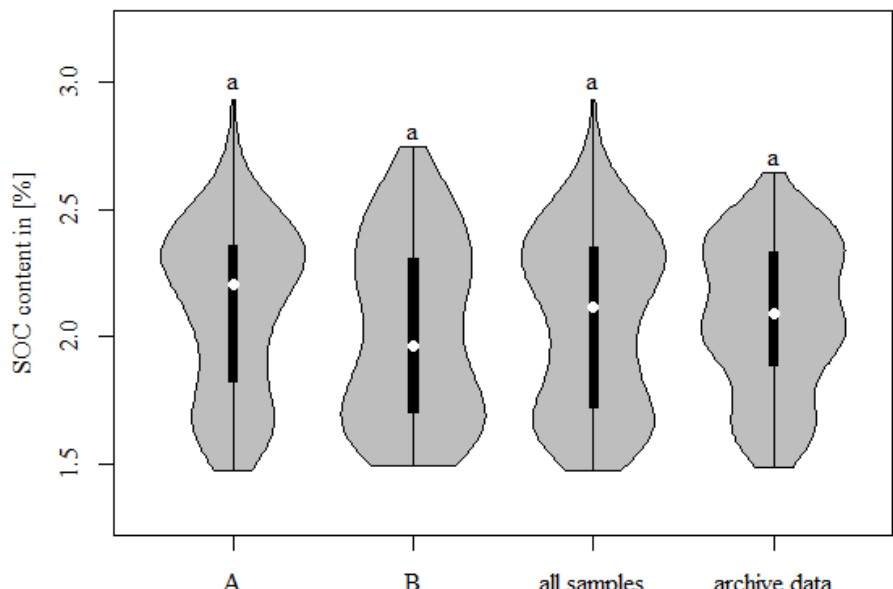

**Fig. 7 Soil organic carbon (SOC) content of the three soil sample sets "A" (left), "B" (middle), and all (middle) and of archive data measured from 2004 to 2007 (right); The thin line shows the 95 % confidence interval, the bar the interquartile range and the dot the median; Mann–Whitney U test was used to compare "A", "B", and all samples to the archive data; the three soil sample sets were not compared among each other.**

**Table 2 Statistics of soil organic carbon in [%] for the three different soil sample sets and the per-plot soil archive**
**data.**

| Samples | Min. | 1st Qu. | Median | Mean | 3rd Qu. | Max. |
|---|---|---|---|---|---|---|
| "A" | 1.47 | 1.82 | 2.21 | 2.11 | 2.36 | 2.93 |
| "B" | 1.49 | 1.70 | 1.97 | 2.02 | 2.31 | 2.74 |
| all | 1.47 | 1.72 | 2.12 | 2.01 | 2.35 | 2.93 |
| Archive data | 1.49 | 1.89 | 2.09 | 2.08 | 2.33 | 2.64 |

### 3.2    Comparison of data sets and pre-processing methods

Table 3 displays the different combinations of soil spectra and SOC values per soil sample. The resulting models are then named after following scheme: Modelset$_{x1\ x2\ x3}$ with the SOC measurement error (x1), the spectral



measurement error (x2) and the within-sample variability (x3). The number 1 indicates that the respective error

is included into the model, the number 0 shows that the error was removed beforehand by averaging the data.

**Table 3 Data basis per soil sample: Number of SOC values, number of measured spectra and resulting spectra per soil sample are shown.**

|  | Number of SOC values per sample | Number of measured spectra per sample | Resulting spectra per sample for model building |
|---|---|---|---|
| **Modelset$_{111}$** | 3 | 18 | 54 |
| **Modelset$_{101}$** | 3 | 6 | 18 |
| **Modelset$_{100}$** | 3 | 1 | 3 |
| **Modelset$_{011}$** | 1 | 18 | 18 |
| **Modelset$_{001}$** | 1 | 6 | 6 |
| **Modelset$_{000}$** | 1 | 1 | 1 |

Fig. 8 shows boxplots of the RMSE$_{MV}$ obtained with the data sets. The six model sets are displayed in one panel.

The models using "A" samples are shown in the 1$^{st}$ column (a), "B" samples in the 2$^{nd}$ column (b) and all

samples in the 3$^{rd}$ column (c); the numbers 1 to 4 refer to the used pre-processing method. As 10-fold CV with

five repetitions was performed, five RMSE$_{MV}$ are shown for each data set. The model results are now compared

based on their mean RMSE$_{MV}$ and their interquartile range. Comparing Modelset$_{111}$ with Modelset$_{000}$ shows how

the inclusion of all input data uncertainties affects the model results. It can be seen that a model without error

propagation (Modelset$_{000}$) reaches a RMSE$_{MV}$ 0.12 - 0.13 % SOC and R$^2$ 0.85-0.87 using the pre-processing

method which delivered the best results. A model with error propagation (Modelset$_{111}$) on the other hand reaches

a RMSE$_{MV}$ 0.16 - 0.17 % SOC and R$^2$ 0.76 - 0.78. In general, Modelset$_{000}$ leads to better predictions with only

two exceptions (a3 and b3). This is further illustrated in Fig. 9**Fehler! Verweisquelle konnte nicht gefunden**

**werden.** and could be expected, as Modelset$_{000}$ contains no uncertainties which could influence the model

outcome. What happens when only some of the input data uncertainties are implemented shows the

consideration of Modelset$_{001}$ and Modelset$_{101}$. Modelset$_{001}$, including only the within-sample variability, led to

better results (R$^2$ 0.81 – 0.82, RMSE$_{MV}$ 0.14 – 0.15 % SOC) than Modelset$_{101}$ (R$^2$ 0.72 – 0.75, RMSE$_{MV}$ 0.17 –

0.18 % SOC) which includes both SOC measurement error and within-sample variability. Comparing

Modelset$_{001}$ to Modelset$_{000}$ shows that the influence of the within-sample variability alone leads almost always to

slightly poorer model results. Modelset$_{100}$, containing only the SOC measurement error, delivers in most cases

distinct poorer results than Modelset$_{000}$. This indicates that the measurement error implemented in the SOC

content measurements has a high effect on the model outcome if the data basis, in this case the available spectra





per soil sample, is small. When the data basis per sample is higher, as for example for $Modelset_{011}$ and $Modelset_{111}$ (see Table 3), the model may be able to process the information contained in the data set in a better

way, leading to nearly similar results for $Modelset_{011}$ and $Modelset_{111}$ ($R^2$ 0.77 – 0.79, $RMSE_{MV}$ 0.15 - 0.16 and $R^2$ 0.76 – 0.78, $RMSE_{MV}$ = 0.16 – 0.17). These results show the high influence the treatment of the input data has on the model outcome and on the error measures.

The combination of the pre-processing methods SG and CR was found to be the best pre-processing variant for all models built with the "A" samples (a1 to a4) and also for the models built with all data (c1 to c4) with the

exception of $Modelset_{100}$. For the models built with the "B" samples, no clear best pre-processing method could be recognized. The models in b1 to b4 though show a similar response to the pre-processing methods with SNVd1 leading to the poorest model results in almost every case.

Regarding only the models derived from "A" and "B" samples (a1 to a4 and b1 to b4) it becomes clear that the models using "B" samples led to better predictions than those using "A" samples. The locations of the "B"

samples were determined using the Kennard-Stone-algorithm, those of the "A" samples with k-means clustering algorithm. Fig. 7 allows an assessment of the quality of those two sampling designs and shows no clear resemblance between the violin plots of "A", "B", "all samples" and the archive violin plot. The Mann–Whitney U test did not show a significant difference between the archive data and the sample sets used in this study. "A" samples as well as "B" samples seem to represent the LTFE SOC data in an adequate way. As already mentioned

above, the difference in the distribution of SOC values of "A" and "B" samples might have led to a different predictive capability in certain SOC value ranges. If this difference is the reason for the better performance of the "B" models cannot be stated with certainty.

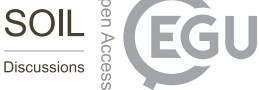

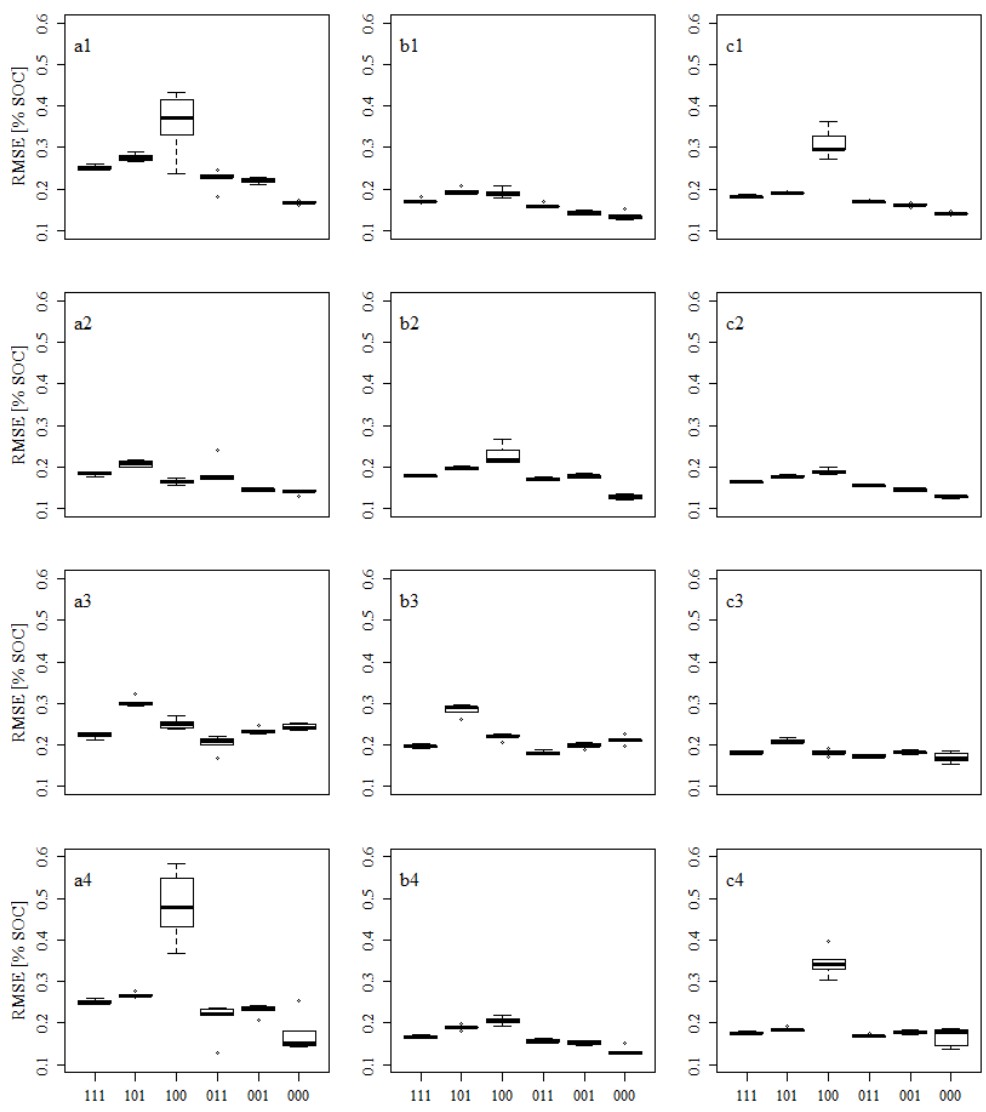

**Fig. 8 Results obtained with the data sets; Figure columns refer to data sets using a) "A" samples, b) "B" samples and**

**c) all samples; figure rows refer to the applied pre-processing, 1 = SG, 2 = SGCR, 3 = SNVd1, 4 = gapDer.**



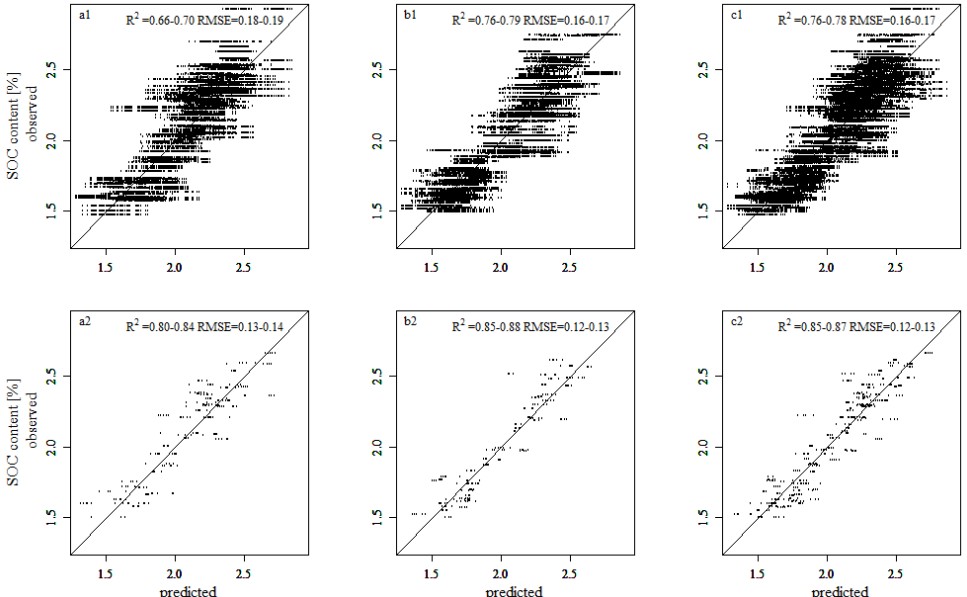

**Fig. 9 Comparison of predicted and observed soil organic carbon (SOC) values for Modelset$_{111}$ (a1 to c1) and Modelset$_{000}$ (a2 to c2) for five repetitions with pre-processing SGCR; a) shows results for "A" samples, b) for "B" samples and c) for all samples.**

Examples of other studies using Vis-NIR spectra to predict SOC are listed in Table 4. Most studies used a different number of scans to get an averaged spectrum to predict SOC, so the error implemented in the respective input data is assumed to be different. As shown in section 3.1, the input data error has a major influence on the model outcome. In none of these studies the error in SOC measurements is mentioned to be considered during model building. Also, in most studies the available data set is randomly parted into calibration and validation set, using different percentages of the data for the two sets. Jeong et al., 2017 showed that different validation strategies lead to different error measures.

The overall best pre-processing method in this study was the combination of SG and CR. SG was used successfully by many authors before for spectral pre-processing (Bogrekci and Lee, 2006; Nocita et al., 2013; Stevens et al., 2013; Viscarra Rossel et al., 2006a). CR was used by e.g. Viscarra Rossel et al. (2016) or Loum et al. (2016) with acceptable success. The combination of SG and CR could not be found in literature. SNV was applied before by other authors in order to remove baseline effects (Knadel et al., 2015; Minasny et al., 2011; Viscarra Rossel et al., 2006a). The pre-processing technique d1 was often found to lead to poorer model results




in this study. This may have its cause in the tendency of d1 to over-fitting and the increasing of noise in the data

as reported by some authors before (Leone et al., 2012; Stevens and Ramirez Lopez, 2014). Leone et al. (2012) suggests the usage of SG in combination with d1 to solve the problem. For the usage of gapDer no comparison could be found in literature. As there is no standard pre-processing technique which works on all spectral data (Stenberg and Viscarra Rossel, 2010), it is recommended to always test various techniques and to choose the one which performs best for the respective data.

**Table 4 $R^2$ from literature for soil organic carbon prediction models.**

| Author | number of samples | averaged scans per sample | calibration and validation set | $R^2$ |
|---|---|---|---|---|
| (Reeves and Smith, 2009) | 720 | 64 | a) Cross-validation with all samples<br>b) Independent validation set | a) 0.534<br>b) 0.335 |
| (Islam et al., 2003) | 161 | - | Randomly selected from data set (121 / 40) | 0.76 |
| (Wang et al., 2014) | 156 | 4 | Randomly selected from data set (116 / 40) | 0.67 - 0.88 |
| (Volkan Bilgili et al., 2010) | 512 | 100 | Randomly selected from data set (70 % and 30 %) | 0.80 |
| (Kuang and Mouazen, 2013) | 174 | 10 | 60 % and 40 % | - |
| (Jiang et al., 2016) | 98 | 10 | Data set parted into calibration and validation set | 0.58 - 0.85 |
| (Conforti et al., 2015) | 201 | 30 | - | - |
| (Leone et al., 2012) | 374 | 4 | Randomly selected from data set (2/3 and 1/3) | 0.84 - 0.92 |

## 4    Conclusions

This study aimed to investigate the influence different aspects have on the model building process and the calculation of error measures. Those aspects included the input data uncertainties, the number of measurements per sample and the chosen pre-processing method. Furthermore, the effect of sampling design, model tuning and

validation procedure was discussed. The influence of the measurement error for both SOC and spectral measurements was apparent in this study, leading almost always to worse model results. As the usage of different devices will lead to different measurement errors, other authors (Ben Dor et al., 2015; Pimstein et al.,

2011) recommend the application of an internal reference to compare the measurements taken from different

devices. The need of such an internal reference for spectral measurements is illustrated by the results of this

study. Although the samples were sieved and homogenised before taking measurements, within-sample

variability sill had an influence on the model outcome. In general, the variability of the samples depends on the

soil treatment and possibly also on the origin of the samples (e.g. agricultural or forest soils). The two sampling

designs used in this study were tested on their statistical inference. It became apparent that both were

representative of the SOC values existing in the LTFE. Nevertheless they led to models with different predictive

capability, showing the influence the chosen sampling design has on the outcome. The impact of the different

pre-processing methods showed clearly in this study, but it may be different when using other data sets, though.

Therefore, various pre-processing techniques should be tested and compared beforehand. All in all, this study

showed that it is important to clarify which information the error measure contains for a certain model. In order

to compare the error measures obtained with different model approaches, several points have to be considered.

These points include the number of sub-samples and measurements taken for one sample and the sample rotation

made during measurement. It also has to be made certain that autocorrelation between calibration and validation

sets is avoided during the model building process.

**Acknowledgements**

We are grateful to the support of our colleagues from the UFZ departments Soil System Science, Community

Ecology, Monitoring & Exploration Technologies, and Computational Landscape Ecology for support during the

field campaign and the consequent sample preparation and lab analysis.

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
