# Peer review of "Error propagation in spectrometric functions of soil organic carbon"

_SOIL, 2018_

## Referee Comment (RC2)

**mythbusters in chemometrics**
doi: 10.1255/nirn.1438

**Myth: A partial least squares calibration model can never be more precise than the reference method...**

[Figure]

Guest-buster: Peter Paash Mortensen
**Ms.Chem, Ph.D. in PAT and process sampling**

The Mythbuster column is always watching out for myths, old or new... In this column we get help from a friend and colleague, Peter Paasch Mortensen, who is an experienced industrial chemometrician. Peter has worked for 15 years at Novozymes (Denmark) and is now with Arla Foods (Denmark).

This myth usually crops up in the wide field of *practical chemometrics,* when we care to validate a given multivariate calibration model. Here we skip our usual grumpy comments re *proper* validation; we shall here picture a calibrator who has accepted the basic principles of validation, which will then be based on independent test sets (test set validation).

The Mythbuster column has previously displayed control charts tracking total measurement uncertainties for a secondary method, in this case near infrared (NIR) spectroscopy. In the applied chemometrics world, it is often implicitly assumed that differences between predicted NIR results and reference values are mainly associated with uncertainties in the NIR prediction. However, such differences are always differences between two values, each with many error sources, which collectively add up to the total observed deviations. This always leaves the user with the question: which contains the major uncertainty, the NIR prediction or the reference measurements? Only by basing our analysis on solid statistical descriptions of both series and their origin (especially with respect to all measurement errors involved, i.e. sampling, processing, calibration, prediction etc.) can we allow ourselves to make firm conclusions that, for example, may involve changing the state of an industrial production process. Who would want to do such a thing on anything less than well-documented evidence?

Primary analytical methods are commonly accepted as solid evidence but how often do we challenge this general assumption? The validity of a primary analytical method must mean, among other things, that it is under full statistical control, which again means that it is associated with quantified systematic as well as random uncertainty

estimates. It is often *assumed*, with very weak theoretical foundation, however, that systematic errors are negligible for a primary analytical method. But this flies in the face of all established experience in the analytical laboratory. There is always a systematic bias, the question of course is... is the bias small enough to be acceptable?

In this column we show that even severe uncertainty does not harm a partial least squares (PLS) model as long as it is random in nature. But if the primary method contains systematic error effects (analytical bias), or worse, if these are not stable (sampling bias), any attempt to calibrate and validate will suffer a breakdown and become unreliable. Assuming that an analytical bias has been brought under control, there remains the prediction precision which most certainly can be enough of a problem in itself. Although always a potential killer, we shall here leave out the specific sampling issues (much covered elsewhere), except by stating categorically that they are always to be ignored entirely at your own peril.[1]

To establish proof that a primary analytical method is free from bias, it must be specifically tested against well-defined standard materials whereby it can ultimately be related to appropriate international or national standards through an unbroken chain of comparisons (metrological validation). Even in this case, the uncertainty associated with estimation of the reference method bias remains an essential component of the overall measurement uncertainty. We here bring a mainly visual illustration.

This problem is quite simple to *simulate*. In Figure 1, Excel was used to generate two series of analytical results, both with an average concentration = 42, with a coefficient of variation (CV%) = 1 between the labs, which can be considered quite good:

$$STD = 0.418 => CV\% = 0.418 / 42 \times 100\% \sim 1\%.$$

These numbers are meant to illustrate repeated analytical results generated over a

[Figure]

**Figure 1.** Blue lab and red lab analytical result series, based on duplicate samples extracted for control purposes. A bias of 0.5 has been added to the red lab results in the summertime (green period). Standard error of estimate (*SEE*) between labs (unbiased) is 0.32 but during the green sub-period it increases to 0.42. This is barely enough to detect the bias in most cases.

[Figure]

Figure 2. Adding increasing levels of symmetric noise to the same basic PLS model.

products was split into a calibration set of 114 samples; 22 samples were picked from a production run by stratified random selection to form the test set for the present simulations. The basic PLS model was based on full spectra [X] with an appropriate complexity (three PLS-components, no outliers present); the same basic model was used for all noise-added simulations below.

Figure 3 shows the calibration evaluations for the four models illustrated above in convenient predicted vs reference formats. For the specific comparison purpose needed here, one may use root mean square standard error of cross validation (*RMSECV*) [or one could use root mean standard error of prediction (*RMSEP*) based on the test—both options are illustrated here; Figures 3 and 4].

Adding noise to the Y data clearly diminishes the prediction performance of the

year stemming from two labs (red lab and blue lab). Duplicate samples are extracted every two weeks and both labs are then supposedly analysing the *same sample* (barring sampling issues). In the present simulations, a small bias of 0.5 to the results of the red lab has been added during the "green period". There is a very high possibility that this temporary bias would go unnoticed—it is here supposed to emulate a temporary aberrance for the blue lab during summer time.

Detecting a significant bias for a reference laboratory is not easy, but its impact on production, calibration and validation is easy to imagine. If we accept the possibility of a non-zero bias, this will lead to more careful control procedure interpretations when a mismatch between lab and NIR predictions is observed—i.e. it may in fact not only be NIR having the problem.

Another issue is repeatability, i.e. the ability to analyse/predict the same result repeatedly within an acceptable narrow range. It will often be the case that a NIR prediction method actually has better repeatability than the relevant primary laboratory reference methods, e.g. traditional wet chemistry methods. Can a NIR PLS model actually perform better in repeatability than the relevant primary method?

Let us build an instructive PLS model (X = NIR) with Y-data displaying zero bias but with various levels of noise added.

**Experiment and simulations**

The question at hand is how the intrinsic reference laboratory uncertainty will affect predictions and validation using PLS modeling. It was decided to test the Myth: "A calibration can never be more precise than the reference analysis" using realistic *simulation* by adding various, non-trivial levels of noise to the unbiased Y-data at levels of

2%, 5% and 10% *CV*, respectively. (A later Mythbuster column will illustrate adding identical noise levels as above but to distinctly *biased* Y-data.)

One hundred and thirty-six (136) data pairs (FT-IR spectra) of various pharmaceutical

[Figure]

Figure 3. PLS predictions showing predicted vs reference values for the four models in Figure 2.

[Figure]

Figure 4. Prediction of the independent test set (22 samples) using the same four models depicted in Figure 2. Test set validation shows that *RMSEP* only increases from 0.24 to 0.38—expressed in the same units as the *RMSECV*.

model. *RMSECV* ranges from 0.30 (basic model, no noise added yet) to 4.36 for the highest noise addition (10%). Significant noise additions are corrupting the possibilities for detailed prediction precision.

The same four models were subsequently also tested on the independent test set (22 samples), with results shown in Figure 4.

The remarkable effect is that all model performances are now almost identical. The reason is obvious: the calibration consists of 114 data pairs (X-spectrum and corresponding Y-reference value) which tends to *average out* the added random analytical errors.

**Conclusion**

As long as the PLS model is built on data with an acceptably small analytical bias, prediction precision is actually smaller than that of a noise-prone Y-reference method. It may often provide robust and acceptable results even when the reference method is fraught with huge random fluctuations.

The take-home message from the guest buster is that PLS calibration is not sensitive to high laboratory uncertainties (random measurement noise). PLS models with reasonably strong X–Y correlation, to which is added *symmetric noise* (i.e. unbiased noise), reveal that this type of noise stabilises model building and results in less prediction uncertainty than that which characterises the Y-reference uncertainties themselves.

Myth busted!

P.S. A later column shall address the identical issue when played out against the background of both an analytical bias (*constant*) and one reflecting an *inconstant* bias (sampling bias). Readers can find a sneak preview of problems surrounding measurement error effects with and *without* proper attention to specific sampling issues in Reference 2.

**References**

1. DS 3077 (2013) DS 3077 Representative sampling—Horizontal Standard. Danish Standards. http://webshop.ds.dk/Files/Files/Products/M278012_attachPV.pdf

2. K.H. Esbensen and C. Wagner, "Theory of sampling (TOS) versus measurement uncertainty (MU)—A call for integration", *TrAC Trends Anal. Chem.* **57,** 93–106 (2014). doi: http://dx.doi.org/10.1016/j.trac.2014.02.007

---

## Referee Comment (RC1) · Anonymous Referee #1 · 28 Feb 2019

Review manuscript EGU SOIL; Error propagation in spectrometric functions of soil organic carbon The aim of the study was to evaluate visible and near infrared (VIS-NIR) reflectance spectroscopy to predict SOC and particularly the source of errors associated. The error propagation when assessing SOC with VIS-NIR has been overlooked in previous studies using that technique. This study is relevant because the precise monitoring of SOC with conventional methods is labour intensive and expensive. In my opinion: - The study is original and in line with EGU SOIL topics. - The manuscript is well organized and has a good study design. - This study is a sufficient contribution to knowledge to be worth publishing in EGU SOIL. - All the sections of the MS are complete, well written and interesting to read (the manuscript has apparently passed through a round of peer-reviewing).

[Figure]

However, VIS-NIR model building process and its associated calculation of uncertainty is not directly in my line of expertise. I recommend to find other peer reviewers. Specific comment: Lines 293-294 "... Fehler! Verweisquelle ... warden" this entence should be delete or translated in English. I have no other specific comment.

Please also note the supplement to this comment:
https://www.soil-discuss.net/soil-2018-42/soil-2018-42-RC1-supplement.pdf

―――――――――――――――――

---

## Referee Comment (RC3) · Anonymous Referee #3 · 20 Mar 2019

**General comments**

The study by Ellinger et al. represents a under-researched topic with high practical relevance for soil spectroscopy applications. An exhaustive set of uncertainty factors contributing on the model error was examined, using a full factorial design. The chosen title is attractive and is well aligned the objective and the performed statistical analyses. Statistical model development and assessment was done using a state-of-the-art cross-validation technique. The assessment of uncertainty comprises a combination of two sampling strategies including their combination, four spectral preprocessing methods, and spectral and analytical data set combinations with different degree of averaging random noise.

This discussion article is well structured and describes the technical aspects in precise and easy understandable plain language. Although covering rather fundamental spectroscopy modeling research, the spectroscopic calibration of samples from a long fertilization trial embeds the uncertainty analysis into a realistic and interesting application context (soil monitoring). This study backs up the current practice of measuring several sample or sub-sample replicate and subsequent spectral averaging, which can improve model performance in many cases. Further, the data set and the error analysis is of particular interest for the soil science community because it quantifies the contribution of random analytical reference method errors on spectral predictions. This paper would comprise an even more valuable scientific contribution if the authors elaborated more on this particular uncertainty component.

The inclusion of the sampling design as a factor together with the other varying uncertainty components requests more concentration from the reader to understand the experiment and read the results. In order to simplify the results and to make the message more concise, the authors could focus on the scenario with the combined "A" and "B" sampling strategies, and move the results of "A" and "B" alone to the appendix.

The model tuning (number of PLSR components) results would be important for the interpretation of model performance under the uncertainty scenarios, but these are missing. These together with a more detailed discussion of error patterns not following the general trend or expectations would help to validate and explain the highly variable results. Further, light scattering effects as important source of spectral model error, and their dependence on soil composition and texture could be discussed, taking into account the applied spectral preprocessing methods. Addressing the specific comments, this paper will be a scientifically sound and valuable soil spectroscopy case study.

**Specific comments**

The suggested references are listed in the bottom of this review.

*Abstract*

Instead of giving a general description of model building, more specific details on the error propagation experiment would show the importance and the quality of the carefully conducted statistical experiment. The abstract lacks quantitative results about the contribution of tested uncertainty factors on model performance.

*Introduction*

The introduction reads fluently and is of adequate length.

The importance of soil organic carbon for soil functioning and frequent measurements provides a good motivation to introduce the methodological relevance of the present study. The reader of this journal is likely already familiar with the role and importance of soil organic carbon an its assessment in soils; therefore, this part can be condensed.

There is some statements that can be assumed common knowledge. As an example, the following sentence in l. 42–44 requires no references: "The precise monitoring of SOC on a LTFE with conventional lab analysis is labour-intensive and expensive (Adamchuk and Viscarra Rossel, 2010; Loum et al., 2016) as it requires the analysis of a rather high amount of samples."

The authors clearly state the motivation of using soil spectroscopy. Shortening the paragraph on soil organic carbon, there is space to briefly explain principles that enable sensing of carbon by infrared spectroscopy (e.g. relationship between functional groups in soil constituents, absorption of electromagnetic radiation and vibration of

bonds in the infrared range, and soil properties, and how these relationships were used for statistical modeling).

The theoretical foundation of uncertainty in modeling is rather sparse and short in relation to the general introduction into concepts of soil organic carbon and advantages of spectroscopy (see also comment of anonymous referee No. 2). The authors are advised to add more key concepts and terminology of uncertainty in the context of predictive modeling generally, and soil spectroscopy modeling applications specifically. It is worth mentioning that measurement and prediction errors can be separated into systematic (bias) and random errors (uncertainty, precision). There are several sources of uncertainty in the model predictions, such as predictor (spectra) measurement errors, response (chemical laboratory measurements) errors and model errors (related to model instability in model parameter estimates or model structure). For analytical data, random errors are most relevant. Further, both bias and variance contribute to the uncertainty in generalization error estimate (here RMSE), which is another source of uncertainty. The authors mention and stress importance of resampling strategy, which was comprehensibly explain. To complement, a link to the conditional model error could be made (for example citing Beleites et al. (2005)). Some more details that are more relevant for the authors' experiment can also be added to the existing section in Material and Methods. For diffuse reflection infrared spectroscopy, scattering effects are particularly important spectral noise factors, that are relevant under the chosen experimental setup and the objective. Therefore, they merit particular mention. For a general description of model uncertainty analysis, e.g. Jansen and Michiel (1998) provide a good reference.

l. 55: Wetterlind et al. (2013) recommend general strategies for spectral measurement and modeling; Spectral averaging is also recommend. Therefore, this would be an important message and reference to add to the existing ones. Further, Wetterlind et al. (2013) highlight the effect of the sampled area and advise to perform replicate spectral

sampling for small areas.

*Material and Methods*

The model of the CN analyzer is not described.

l. 104f: "C measurements were taken as organic carbon due to negligibly small carbonate contents." The authors are asked to provide quantitative statement (lower than xxx % C.).

There is no mention of whether or how many scanning replicate measurements were internally averaged (spectrometer setting) prior taking spectrometer readings of the sub-sample and rotation replicate spectra (further noise averaging).

What is the name or composition of the material was used as a white reference (name manufacturer if composition not known)?

The description of the preprocessing techniques is a bit too long and detailed. To keep a focus on the main topic, a brief description in one or two sentences, a citation to the original publication and maybe some soil spectroscopy case study where the respective techniques was applied, suffice. Spectral preprocessing techniques have been well described and researched in chemometrics, and applied in various other scientific discipines and industry over the last decades.

The chosen resampling strategy is particulary well suited when data is scarce and variability is high, but can be recommended in general. This study can serve as an exemplary resampling setup for the soil spectroscopy community (repeated nested group

k-fold cross-validation for model parameter tuning and evaluation). The approach is also particulary consisely described. Repeated 10-fold cross-valiation is a practical and widely used cross-validation strategy to reduce uncertainty in performance estimators. Therefore, adding an other reference to earlier applied predictive modeling literature that study effects of resampling strategy on the estimation of model evaluation metrics is advised. For example, Molinaro et al. (2008) is a suitable reference. The description and the illustration in Fig. 6 refer to a nested or double cross-validation, where the inner resampling layer comprises parameter tuning and the outer layer is used for model evaluation, which is used to avoid selection bias in parameter tuning. The authors should mention the nested or double cross validation terminology and also reference key literature. Varma et al. (2006) is suggested as a reference here.

"Although the samples were sieved and homogenised before taking measurements, within-sample variability sill had an influence on the model outcome. In general, the variability of the samples depends on the soil treatment and possibly also on the origin of the samples (e.g. agricultural or forest soils)." The influence of scattering effects and major importance of texture should be mentioned and backed up with literature in this place.

*Results and Discussion*

*General considerations on the reviewer's comments: Some of the below comments may cast a rather sceptical view on the authors' explanation of the results. However, the experimental conditions (in a statistical sense) are not necessary such that there is major weak points of the results. The authors are kindly asked to provide a response or some more results or explanations as outlined below (appendix). The intention is to challenge the author's hypotheses. There are unfortunately no means to compare these results to other similar studies in the soil spectroscopy literature. Nevertheless,*

[Figure]

*the spectroscopy community has been following the general recommendation to average noise from replicate measurements to obtain good performing and robust models. This study showcases some "surprising effects". There are no consistent patterns that clearly show beneficial effects of averaging and removing spectra under all conditions. Some inconsistent patterns many arise from interaction in contributing factors. A repeated nested 10-fold cross-validation guarantees that parameter selection is unbiased, but is prone to overfitting if sample grouping is only respected in assessment and not in tuning layer, in combination of multiple spectral and/or analytical replicate data set that are used for modeling.*

How many PLSR components were used in the final models? What was/were the frequency/frequencies of respective best number of components across the folds and repeats?

For Savitzky-Golay (method 1) and the Norris gap derivative (method 4), the RMSE in model set 101 is considerably higher than in model set 100 as well as all other sets, whereby 3 times 6 identical spectra instead of 3 times an identical average spectrum are used (Fig. 8). Could it be that this is the result of a deleterious preprocessing effect due to missing sensor jump correction (see Fig. 3)? Noise enhancement may occur under certain preprocessing strategies such as the gap derivative (see also comments in next paragraph for further interaction possibilities). Fig. 5 shows that the spectral variation largely manifests as offset variation. Savitzky-Golay smoothing with no derivative does not remove offset errors. This is worth a discussion. Assuming identical resampling sets among preprocessing methods within a data set, there seems to be a systematic error component in the spectra in data set A for these cases. The authors are encouraged to recompute results after correcting for sensor offsets, or they should at least consider possible explanations for clearly poorer model performance in the discussion. How do the authors explain such a drastic error increase in the data situation from 3 times 6 different averaged spectra to 3 times one final average spectrum

per sample when using 3 analytical replicates? Why does this occur only when adding replicate analytical measurements to the modeling process, but not when averaged analytical measurements are used?

The nested cross-validation can yield different optimal number of component for each fold, which can be justified such that different data situations comprise different optimal model parameters with respect to performance. The nested or double cross-validation scheme in Fig. 6. shows the model validation procedure in the outer cross-validation loop within the left partition set and the model tuning resampling procedure within the right partition set. The authors mention that k-fold cross-validation was done by assigning the entire set of replicate spectra of respective samples in either fitting or hold-out sets. The authors don't explicitly state whether this grouping by sample is done for the both model evaluation k-fold cross-validation layer and model tuning cross-validation layer, or just in either one of them. This lets room for ambiguous interpretations of the experiment and confounding factors. The R function 'tuneControl' used for the tuning resampling, but the 'groupKFold()' function from the caret RÂăpackage – which splits data based on a grouping factor – is not mentioned. Namely, focusing on the results depicted in panel "c1"Âăof Fig. 8, the prediction error decreases when less sub-sample replicate spectra are included in the modeling.
Assuming that simple 10-fold cross-validation without grouping, the tuning layer would suffer from a data leakage from fitting folds into the respective assessment folds when replicate spectra are present. Data leakage provides biased tuning results and can select very high number of components. The maximum number of PLSRÂăcomponents was set 40, which represents severe over-fit with 100 rows and so many predictors ("large $p$ small $n$"" problem). The reader cannot interpret if there was variability of the number of finally selected components for the different resampling sets. The authors should therefore provide a $ncomp$ final tuning results table in the appendix if they were variable among the sampling sampled data set, model set and preprocessing method combinations. In worst case scenario where the outer assessment resampling is grouped, but the inner tuning resampling lacked sample grouping, presence of more replicate spectral measurements per sample can yield too high $ncomp$ and too adaptive models, This would give an alternative explanation for poor performance in the outer assessment when all replicate spectra are used for modelling, in comparison to model set011 and set001 with increasing degree of spectral averaging and less likely too adaptive PLSRÂămodels. Using the full set of analytical carbon measurement replicates can have generally lower performance. Directly addressing these aspects is an outstanding achivement of this study. The authors are highly encouraged to extend the discussion about effects of analytical uncertainty, taking into account the considerations of anonymous referee No. 2.. To sum up, the authors are kindly asked elaborate and comment on interactive effects between the sources of errors and the resampling, thereby also critically address confounding effects as hypothesized above in the author's response to the review. Did the authors implement the grouped k-fold procecedure for both inner an outer cross-validation layers?

The authors could move Fig. 3 to the results and further illustrate differences in preprocessed spectra for replicate spectral measurements (e.g. one example spectrum). Further, texture might explain different model performances for data sets "A"Âăand "B". Sandy soils are known to confound increases in reflectance typically also found for increases in soil organic C (see e.g. Stevens et al., 2013).

The use of different preprocessing techniques has already been exhaustively discussed in the soil spectroscopy literature. The performance of different preprocessing methods depends on the data context, as stated in the last sentence of paragraph l. 347–349. Thus, such a comparison to other studies does not make sense and brings no added value to the reader. The authors are advised to remove the comparison. To mention that the effects of signal processing methods and associated parameters on model performance are study and data specific is sufficient (incl. references).

*Conclusion*

"Autocorrelation between calibration and validation sets". It is valuable for the soil spectroscopy community that the present study considers and stresses data grouping effects in resampling, here repeated measures, and accounts for those in the cross-validation procedure. Ignoring such grouping factors can result in over-optimistic estimation of the model performance due to leakage of predictive relationships from the modeling into assessment sets. The scientific community would benefit if authors could name the strategy using the terminology "group k-fold cross-validation".

**Technical corrections**

l. 86–89: "Categorical and continuous data first entered a factor analysis with mixed data (FAMD) performed with R package FactoMineR (Lê et al., 2008) to allow for further joint analysis. For design 'A' the LTFE plots were then grouped by a k-means cluster analysis. R package NbClust (Charrad et al., 2014) automatically determines the optimal number of clusters making use of 30 indices." How many factors from previous factor analysis with mixed data are retained and used for k-means clustering? How much variance do these factors and continous variables explain.

l. 127–132: The outlier analysis is in section 2.4, but is not considered as preprocessing. This part needs either a separate section or a generic data analysis section.

l. 263f: "The violin plots of all three soil sample sets do not resembles the archive violin plot very much." typo: "resemble"

l. 344: "The simple first derivative (d1) performs poorly because it increases noise as stated, but there is no tendency to overfit related to this particular preprocessing tech-

nique (rather consequence of improper model resampling and tuning setup in combination with adaptive models)"; typo: "poorly" .

l. 365f: "The impact of the different pre-processing methods showed clearly in this study, but it may be different when using other data sets, though." This sentence needs to be more precise; the authors are kindly advised to use "impact the different pre-processing methods on model performance..." or mention variable performance results or similar.

l. 239f: "In this study, the partition of the spectral data into 10 folds for 10-fold CV had to be done very carefully, as in some cases multiple spectra existed for one sample.' Expression "very carefully" needs to be rephrased in scientific language, as well as "some cases" (2/3 of the cases).

*Figures*

General comment for panel figures: Experimental labels are hidden within the plot area and are better placed on top left of each subplot to facilitate the reader's visual perception.

Fig. 3: The spectra have not been joined correctly in the sensor jump region (900nm). On the left side of the sensor shift there is a small peak which appears to be identical on the right side.

Fig. 5: Zooming into the range with highest replicate spectral variation would help to discriminate single replicate spectra and help the reader to visually assess the type of errors in the spectra (random noise vs. systematic offset).

Fig. 9: The figure needs quality needs to be improved for print. Plots should be in a vector format. Increase the font sizes for axis labels. Also, mathematical expressions should be separated by full or half spacing. Use minus sign or alternatively en dash instead of hyphen for minus. R-squared and RMSEÂăcould be placed below each other for better readability. Hollow circles or transparency, and bigger point symbols to deal with overplotting are recommended. Letters "a/b" in panel labels should be capitalized because the sampling designs are abbreviated as "A/B" elsewhere.

**References**

Beleites, C., Baumgartner, R., Bowman, C., Somorjai, R., Steiner, G., Salzer, R., & Sowa, M. G. (2005). Variance reduction in estimating classification error using sparse datasets. Chemometrics and Intelligent Laboratory Systems, 79(1–2), 91–100. https://doi.org/10.1016/j.chemolab.2005.04.008

Jansen, M. J. W. (1998). Prediction error through modelling concepts and uncertainty from basic data. Nutrient Cycling in Agroecosystems, 50(1), 247–253. https://doi.org/10.1023/A:1009748529970

Molinaro, A. M., Simon, R., & Pfeiffer, R. M. (2005). Prediction error estimation: a comparison of resampling methods. Bioinformatics, 21(15), 3301–3307. https://doi.org/10.1093/bioinformatics/bti499

Stevens, A., Nocita, M., Tóth, G., Montanarella, L., & van Wesemael, B. (2013). Prediction of Soil Organic Carbon at the European Scale by Visible and Near InfraRed Reflectance Spectroscopy. PLoS ONE, 8(6), e66409. https://doi.org/10.1371/journal.pone.0066409
Varma, S., & Simon, R. (2006). Bias in error estimation when using cross-validation for model selection. BMC Bioinformatics, 7, 91. https://doi.org/10.1186/1471-2105-7-91

Wetterlind, J., Stenberg, B., & Rossel, R. A. V. (2013). Soil Analysis Using Visible and Near Infrared Spectroscopy. In F. J. M. Maathuis (Ed.), Plant Mineral Nutrients (pp. 95–107). Totowa, NJ: Humana Press. https://doi.org/10.1007/978-1-62703-152-3_6
* * *

---

## Author Comment (AC1) · 21 May 2019

Dear reviewer

We thank you for your time and valuable comments. Please find our replies displayed in blue color below each comment.

Kind regards
Mareike Ließ

**Anonymous Referee #1**

Review manuscript EGU SOIL; Error propagation in spectrometric functions of soil organic carbon The aim of the study was to evaluate visible and near infrared (VIS-NIR) reflectance spectroscopy to predict SOC and particularly the source of errors associated. The error propagation when assessing SOC with VIS-NIR has been overlooked in previous studies using that technique. This study is relevant because the precise monitoring of SOC with conventional methods is labour intensive and expensive. In my opinion: - The study is original and in line with EGU SOIL topics. - The manuscript is well organized and has a good study design. - This study is a sufficient contribution to knowledge to be worth publishing in EGU SOIL. - All the sections of the MS are complete, well written and interesting to read (the manuscript has apparently passed through a round of peer-reviewing). However, VIS-NIR model building process and its associated calculation of uncertainty is not directly in my line of expertise. I recommend to find other peer reviewers. Specific comment: Lines 293-294 ": : : Fehler! Verweisquelle : : : warden" this entence should be delete or translated in English. I have no other specific comment.
Reply: Adapted accordingly.

---

## Author Comment (AC2) · 21 May 2019

Dear reviewer

We thank you for your time and valuable comments. Please find our replies displayed in blue color below each comment.

Kind regards
Mareike Ließ

**Anonymous Referee #2**

This study addressed a very important issue in soil spectroscopy field. The data uncertainty is a major factor to affect calibration process. I have some general comments on this study:
1. In the introduction part, it needs more literature reviews on the data uncertainty effect on the modelling process, to enhance awareness of this issue, especially soil data uncertainty from lab chemistry analysis. Because so far not many people even noticed about it, it has been often ignored.
2. For the experimental design of this study, i suggest author should focus on effect of data uncertainty from lab chemistry analysis, how this uncertainty can effect calibration and validation. Because this is the most important part of the issue.
Reply: We have included more references on uncertainty and emphasized the various sources of uncertainty and their impact on model performance.

Consequently, authors only mentioned about this effect a bit in results and discussion part. It would be a very important study in soil spectroscopy community if author could deeply discuss this effect. Because most research project does not allow all soil samples to measure 3 replicate, due to budget limitation. Therefore, the dataset in this study is a treasure.
3. I have attached a small but very good discussion note from NIR news, hope authors can get some ideas about this issue (this note is not reviewer's publication).
Reply: Thank you. We have adapted the introduction, results and conclusion section and included a reference to the discussion news.

---

## Author Comment (AC3) · 21 May 2019

Dear reviewer

We thank you for your time and valuable comments. Please find our replies displayed in blue color below each comment.

Kind regards
Mareike Ließ

**Anonymous Referee #3**
**General comments**
The study by Ellinger et al. represents a under-researched topic with high practical relevance for soil spectroscopy applications. An exhaustive set of uncertainty factors contributing on the model error was examined, using a full factorial design. The chosen title is attractive and is well aligned the objective and the performed statistical analyses. Statistical model development and assessment was done using a state-of-the-art cross-validation technique. The assessment of uncertainty comprises a combination of two sampling strategies including their combination, four spectral preprocessing methods, and spectral and analytical data set combinations with different degree of averaging random noise. This discussion article is well structured and describes the technical aspects in precise and easy understandable plain language. Although covering rather fundamental spectroscopy modeling research, the spectroscopic calibration of samples from a long fertilization trial embeds the uncertainty analysis into a realistic and interesting application context (soil monitoring). This study backs up the current practice of measuring several sample or sub-sample replicate and subsequent spectral averaging, which can improve model performance in many cases.
Reply: Thank you.

Further, the data set and the error analysis is of particular interest for the soil science community because it quantifies the contribution of random analytical reference method errors on spectral predictions. This paper would comprise an even more valuable scientific contribution if the authors elaborated more on this particular uncertainty component.
Reply: We have rewritten large sections of the introduction and results section.

The inclusion of the sampling design as a factor together with the other varying uncertainty components requests more concentration from the reader to understand the experiment and read the results. In order to simplify the results and to make the message more concise, the authors could focus on the scenario with the combined "A" and "B" sampling strategies, and move the results of "A" and "B" alone to the appendix.
Reply: We understand the reviewer's concern about the complexity of the paper. However, as the sampling design is an important aspect of this paper. We, therefore, refrain from shifting the corresponding results to the appendix.

The model tuning (number of PLSR components) results would be important for the interpretation of model performance under the uncertainty scenarios, but these are missing. These together with a more detailed discussion of error patterns not following the general trend or expectations would help to validate and explain the highly variable results. Further, light scattering effects as important source of spectral model error, and their dependence on soil composition and texture could be discussed, taking into account the applied spectral preprocessing methods. Addressing the specific comments, this paper will be a scientifically sound and valuable soil spectroscopy case study.
Reply: The PLSR components vary largely in dependence on the pre-processing method. The information on the number of selected components, therefore, did not result very informative. Furthermore, it

distracts the reader from the main message. We, therefore, refrain from including it in this publication. We have adapted the results section concerning the uncertainty scenarios. We have also extended the introduction section to elaborate on the various sources of uncertainty.

**Specific comments**

The suggested references are listed in the bottom of this review.

Abstract

Instead of giving a general description of model building, more specific details on the error propagation experiment would show the importance and the quality of the carefully conducted statistical experiment. The abstract lacks quantitative results about the contribution of tested uncertainty factors on model performance.
Reply: We have rewritten large sections of the introduction and results section to go more into detail on the error propagation experiment. Quantitative results with respect to the contribution of uncertainty factors on model performance were added to the abstract.

Introduction

The introduction reads fluently and is of adequate length. The importance of soil organic carbon for soil functioning and frequent measurements provides a good motivation to introduce the methodological relevance of the present study. The reader of this journal is likely already familiar with the role and importance of soil organic carbon an its assessment in soils; therefore, this part can be condensed. There is some statements that can be assumed common knowledge. As an example, the following sentence in l. 42–44 requires no references: "The precise monitoring of SOC on a LTFE with conventional lab analysis is labour-intensive and expensive (Adamchuk and Viscarra Rossel, 2010; Loum et al., 2016) as it requires the analysis of a rather high amount of samples." The authors clearly state the motivation of using soil spectroscopy. Shortening the paragraph on soil organic carbon, there is space to briefly explain principles that enable sensing of carbon by infrared spectroscopy (e.g. relationship between functional groups in soil constituents, absorption of electromagnetic radiation and vibration of bonds in the infrared range, and soil properties, and how these relationships were used for statistical modeling).
Reply: The section on spectral sensing was moved from the methods section to the introduction. However, we refrain from shortening the section on SOC as it is important to understand the context of the Vis-NIR application: Soil monitoring.

The theoretical foundation of uncertainty in modeling is rather sparse and short in relation to the general introduction into concepts of soil organic carbon and advantages of spectroscopy (see also comment of anonymous referee No. 2). The authors are advised to add more key concepts and terminology of uncertainty in the context of predictive modeling generally, and soil spectroscopy modeling applications specifically. It is worth mentioning that measurement and prediction errors can be separated into systematic (bias) and random errors (uncertainty, precision). There are several sources of uncertainty in the model predictions, such as predictor (spectra) measurement errors, response (chemical laboratory measurements) errors and model errors (related to model instability in model parameter estimates or model structure). For analytical data, random errors are most relevant. Further, both bias and variance contribute to the uncertainty in generalization error estimate (here RMSE), which is another source of uncertainty.

Reply: We have extended the section on the various sources of uncertainty in the introduction. However, as we do not distinguish between systematic and random errors in this study, we refrain from referring to them in this explicit way.

The authors mention and stress importance of resampling strategy, which was comprehensibly explained. To complement, a link to the conditional model error could be made (for example citing Beleites et al. (2005)).
Reply: We have included a reference to Beleites et al. and Molinaro et al. in the context of resampling.

Some more details that are more relevant for the authors' experiment can also be added to the existing section in Material and Methods. For diffuse reflection infrared spectroscopy, scattering effects are particularly important spectral noise factors, that are relevant under the chosen experimental setup and the objective. Therefore, they merit particular mention.
Reply: We have added two references concerning sensor noise and scattering in the introduction section (Schwartz et al., Pilorget et al)

For a general description of model uncertainty analysis, e.g. Jansen and Michiel (1998) provide a good reference.
Reply: Thank you. We added the reference.

l. 55: Wetterlind et al. (2013) recommend general strategies for spectral measurement and modeling; Spectral averaging is also recommend. Therefore, this would be an important message and reference to add to the existing ones. Further, Wetterlind et al. (2013) highlight the effect of the sampled area and advise to perform replicate spectral sampling for small areas.
Reply: We refer to spectral averaging and repeated scans in the introduction and discussion section. We refer to the standard measurement protocol suggested by Pimstein et al. and, therefore, refrain from citing Wetterlind et al. in this context.

Material and Methods

The model of the CN analyzer is not described.
Reply: added

l. 104f: "C measurements were taken as organic carbon due to negligibly small carbonate contents." The authors are asked to provide quantitative statement (lower than xxx % C.).
Reply: Carbonate contents were below detection limit.

There is no mention of whether or how many scanning replicate measurements were internally averaged (spectrometer setting) prior taking spectrometer readings of the sub-sample and rotation replicate spectra (further noise averaging).
What is the name or composition of the material was used as a white reference (name manufacturer if composition not known)?
Reply: The missing information was included

The description of the preprocessing techniques is a bit too long and detailed. To keep a focus on the main topic, a brief description in one or two sentences, a citation to the original publication and maybe some soil spectroscopy case study where the respective techniques was applied, suffice. Spectral preprocessing

techniques have been well described and researched in chemometrics, and applied in various other scientific discipines and industry over the last decades.

Reply: As we also assess the impact of the pre-processing method on the error measure, we think a detailed description is necessary for understandability.

The chosen resampling strategy is particulary well suited when data is scarce and variability is high, but can be recommended in general. This study can serve as an exemplary resampling setup for the soil spectroscopy community (repeated nested group k-fold cross-validation for model parameter tuning and evaluation). The approach is also particulary consisely described. Repeated 10-fold cross-valiation is a practical and widely used cross-validation strategy to reduce uncertainty in performance estimators.Therefore, adding another reference to earlier applied predictive modeling literature that study effects of resampling strategy on the estimation of model evaluation metrics is advised. For example, Molinaro et al. (2008) is a suitable reference. The description and the illustration in Fig. 6 refer to a nested or double cross-validation, where the inner resampling layer comprises parameter tuning and the outer layer is used for model evaluation, which is used to avoid selection bias in parameter tuning. The authors should mention the nested or double cross validation terminology and also reference key literature. Varma et al. (2006) is suggested as a reference here.

Reply: We specified the applied cross validation as nested approach and added Varma and Simon as well as Molinaro et al.

"Although the samples were sieved and homogenised before taking measurements, within-sample variability still had an influence on the model outcome. In general, the variability of the samples depends on the soil treatment and possibly also on the origin of the samples (e.g. agricultural or forest soils)." The influence of scattering effects and major importance of texture should be mentioned and backed up with literature in this place.

Reply: Reference to soil treatment and scattering effects was made in the introduction. We refrain from referring to soil texture as we are at within-field scale and do not have a pronounced textural variability in our dataset. A reference to sample origin is included in the discussion section.

Results and Discussion

General considerations on the reviewer's comments: Some of the below comments may cast a rather sceptical view on the authors' explanation of the results. However, the experimental conditions (in a statistical sense) are not necessary such that there is major weak points of the results. The authors are kindly asked to provide a response or some more results or explanations as outlined below (appendix). The intention is to challenge the author's hypotheses. There are unfortunately no means to compare these results to other similar studies in the soil spectroscopy literature. Nevertheless, the spectroscopy community has been following the general recommendation to average noise from replicate measurements to obtain good performing and robust models. This study showcases some "surprising effects". There are no consistent patterns that clearly show beneficial effects of averaging and removing spectra under all conditions. Some inconsistent patterns many arise from interaction in contributing factors.

Reply: We have adapted the results and discussion section and also refer to these unexpected effects.

A repeated nested 10-fold cross-validation guarantees that parameter selection is unbiased, but is prone to overfitting if sample grouping is only respected in assessment and not in tuning layer, in combination of multiple spectral and/or analytical replicate data set that are used for modeling.

Reply: We have added a corresponding sentence in the conclusions: "We are aware that the consideration of stratified group CV only for model evaluation but not for tuning might impair model performance as suboptimal model parameters might be selected. This will be adapted in future studies."

How many PLSR components were used in the final models? What was/were the frequency/frequencies of respective best number of components across the folds and repeats?
Reply: The PLSR components vary largely in dependence on the pre-processing method. The information on the number of selected components, therefore, did not result very informative. Furthermore, it distracts the reader from the main message. We, therefore, refrain from including it in this publication.

For Savitzky-Golay (method 1) and the Norris gap derivative (method 4), the RMSE in model set 101 is considerably higher than in model set 100 as well as all other sets, whereby 3 times 6 identical spectra instead of 3 times an identical average spectrum are used (Fig. 8). Could it be that this is the result of a deleterious preprocessing effect due to missing sensor jump correction (see Fig. 3)? Noise enhancement may occur under certain preprocessing strategies such as the gap derivative (see also comments in next paragraph for further interaction possibilities).
Reply: We have corrected for the sensor jump and rerun the models. Figures 8 and 9 were adapted accordingly. Still models build on dataset$_{101}$ in some cases perform worse than models build on datsets$_{100}$. We refer to this peculiarity but have no explanation. As nested repeated 5-fold CV resulted in similar model results, we have replaced 10-fold CV by 5-fold to save computation time, the same applies for the number of components tested, which was reduced to 30.

Fig. 5 shows that the spectral variation largely manifests as offset variation. Savitzky-Golay smoothing with no derivative does not remove offset errors. This is worth a discussion. Assuming identical resampling sets among preprocessing methods within a data set, there seems to be a systematic error component in the spectra in data set A for these cases. The authors are encouraged to recompute results after correcting for sensor offsets, or they should at least consider possible explanations for clearly poorer model performance in the discussion.
Reply: We have corrected for the sensor jump and rerun the models.

How do the authors explain such a drastic error increase in the data situation from 3 times 6 different averaged spectra to 3 times one final average spectrum per sample when using 3 analytical replicates? Why does this occur only when adding replicate analytical measurements to the modeling process, but not when averaged analytical measurements are used?
Reply: We have added this aspect in the results section: "It is not surprising that the dataset of 3 SOC replicate measurements with 1 averaged spectrum (Dataset$_{100}$) results in lowest model performance, as the within-sample variance concerning SOC cannot be explained by the contained predictor information, the input data uncertainty propagates through the model building process."

The nested cross-validation can yield different optimal number of component for each fold, which can be justified such that different data situations comprise different optimal model parameters with respect to performance. The nested or double cross-validation scheme in Fig. 6. shows the model validation procedure in the outer cross-validation loop within the left partition set and the model tuning resampling procedure within the right partition set. The authors mention that k-fold cross-validation was done by assigning the entire set of replicate spectra of respective samples in either fitting or hold-out sets. The authors don't explicitly state whether this grouping by sample is done for the both model evaluation k-fold cross-validation layer and model tuning cross-validation layer, or just in either one of them. This lets room for ambiguous interpretations of the experiment and confounding factors. The R function

'tuneControl' used for the tuning resampling, but the 'groupKFold()' function from the caret RˇApackage – which splits data based on a grouping factor – is not mentioned. Namely, focusing on the results depicted in panel "c1"¡aof Fig. 8, the prediction error decreases when less sub-sample replicate spectra are included in the modeling. Assuming that simple 10-fold cross-validation without grouping, the tuning layer would suffer from a data leakage from fitting folds into the respective assessment folds when replicate spectra are present. Data leakage provides biased tuning results and can select very high number of components. The maximum number of PLSRˇacomponents was set 40, which represents severe over-fit with 100 rows and so many predictors ("large p small n"" problem). The reader cannot interpret if there was variability of the number of finally selected components for the different resampling sets. The authors should therefore provide a ncomp final tuning results table in the appendix if they were variable among the sampling sampled data set, model set and preprocessing method combinations. In worst case scenario where the outer assessment resampling is grouped, but the inner tuning resampling lacked sample grouping, presence of more replicate spectral measurements per sample can yield too high ncomp and too adaptive models, This would give an alternative explanation for poor performance in the outer assessment when all replicate spectra are used for modelling, in comparison to model set011 and set001 with increasing degree of spectral averaging and less likely too adaptive PLSRˇamodels.

Reply: Only the outer CV cycle of the nested repeated k-fold CV approach considers stratified group CV (but: spectral replicate measurements and scans per sample were always assigned to the same fold!). This was specified in the corresponding methods section. Of course, considering stratified group CV also for model tuning might improve model performance. As we used the inbuilt function of the Caret package, this was not possible. We are not aware of any publication that actually implemented stratified group CV in parameter tuning. We will, however, implement this in future studies as mentioned in the conclusion section. The PLSR components vary largely in dependence on the pre-processing method. The information on the number of selected components, therefore, did not result very informative. Furthermore, it distracts the reader from the main message. We, therefore, refrain from including it in this publication.

Using the full set of analytical carbon measurement replicates can have generally lower performance. Directly addressing these aspects is an outstanding achivement of this study. The authors are highly encouraged to extend the discussion about effects of analytical uncertainty, taking into account the considerations of anonymous referee No. 2..

Reply: We have further elaborated on this aspect.

To sum up, the authors are kindly asked elaborate and comment on interactive effects between the sources of errors and the resampling, thereby also critically address confounding effects as hypothesized above in the author's response to the review. Did the authors implement the grouped k-fold procecedure for both inner an outer cross-validation layers?
The authors could move Fig. 3 to the results and further illustrate differences in preprocessed spectra for replicate spectral measurements (e.g. one example spectrum).
Further, texture might explain different model performances for data sets "A"¡aand "B". Sandy soils are known to confound increases in reflectance typically also found for increases in soil organic C (see e.g. Stevens et al., 2013).

Reply: These aspects were addressed in our replies to the specific comments.

The use of different preprocessing techniques has already been exhaustively discussed in the soil spectroscopy literature. The performance of different preprocessing methods depends on the data context, as stated in the last sentence of paragraph l. 347–349. Thus, such a comparison to other studies does not make sense and brings no added value to the reader. The authors are advised to remove the

comparison. To mention that the effects of signal processing methods and associated parameters on model performance are study and data specific is sufficient (incl. references).

Reply: We are not aware of any study that actually quantified the effect of spectral pre-processing on model performance and, therefore, refrain from deleting it from our study.

Conclusion

"Autocorrelation between calibration and validation sets". It is valuable for the soil spectroscopy community that the present study considers and stresses data grouping effects in resampling, here repeated measures, and accounts for those in the crossvalidation procedure. Ignoring such grouping factors can result in over-optimistic estimation of the model performance due to leakage of predictive relationships from the modeling into assessment sets. The scientific community would benefit if authors could name the strategy using the terminology "group k-fold cross-validation".

Reply: The aspect was emphasized in the conclusion section.

**Technical corrections**

l. 86–89: "Categorical and continuous data first entered a factor analysis with mixed data (FAMD) performed with R package FactoMineR (Lê et al., 2008) to allow for further joint analysis. For design 'A' the LTFE plots were then grouped by a k-means cluster analysis. R package NbClust (Charrad et al., 2014) automatically determines the optimal number of clusters making use of 30 indices." How many factors from previous factor analysis with mixed data are retained and used for k-means clustering? How much variance do these factors and continous variables explain.

Reply: The FAMD was applied to decorrelate the data. All factors were retained.

l. 127–132: The outlier analysis is in section 2.4, but is not considered as preprocessing. This part needs either a separate section or a generic data analysis section.

Reply: We refrain from including an additional section for only two sentences. As the outlier removal is the first step before applying any spectral pre-processing, we do not see why it should not stay in this section.

l. 263f: "The violin plots of all three soil sample sets do not resembles the archive violin plot very much." typo: "resemble"

Reply: corrected

l. 344: "The simple first derivative (d1) performs poorly because it increases noise as stated, but there is no tendency to overfit related to this particular preprocessing technique (rather consequence of improper model resampling and tuning setup in combination with adaptive models)"; typo: "poorly" .

Reply: corrected

l. 365f: "The impact of the different pre-processing methods showed clearly in this study, but it may be different when using other data sets, though." This sentence needs to be more precise; the authors are kindly advised to use "impact the different preprocessing methods on model performance..." or mention variable performance results or similar.

Reply: The section was rewritten.

l. 239f: "In this study, the partition of the spectral data into 10 folds for 10-fold CV had to be done very carefully, as in some cases multiple spectra existed for one sample.'

Expression "very carefully" needs to be rephrased in scientific language, as well as "some cases" (2/3 of the cases).
Reply: adapted

Figures

General comment for panel figures: Experimental labels are hidden within the plot area and are better placed on top left of each subplot to facilitate the reader's visual perception.
Reply: adapted

Fig. 3: The spectra have not been joined correctly in the sensor jump region (900nm).
On the left side of the sensor shift there is a small peak which appears to be identical on the right side.
Reply: Thank you. We corrected for the sensor jump.

Fig. 5: Zooming into the range with highest replicate spectral variation would help to discriminate single replicate spectra and help the reader to visually assess the type of errors in the spectra (random noise vs. systematic offset).
Reply: Thank you. The figure was adapted, accordingly.

Fig. 9: The figure quality needs to be improved for print. Plots should be in a vector format. Increase the font sizes for axis labels. Also, mathematical expressions should be separated by full or half spacing. Use minus sign or alternatively en dash instead of hyphen for minus. R-squared and RMSE¢acould be placed below each other for better readability. Hollow circles or transparency, and bigger point symbols to deal with overplotting are recommended. Letters "a/b" in panel labels should be capitalized because the sampling designs are abbreviated as "A/B" elsewhere.
Reply: adapted

**References**

Beleites, C., Baumgartner, R., Bowman, C., Somorjai, R., Steiner, G., Salzer, R., & Sowa, M. G. (2005). Variance reduction in estimating classification error using sparse datasets. Chemometrics and Intelligent Laboratory Systems, 79(1–2), 91–100.
https://doi.org/10.1016/j.chemolab.2005.04.008

Jansen, M. J. W. (1998). Prediction error through modelling concepts and uncertainty from basic data. Nutrient Cycling in Agroecosystems, 50(1), 247–253. https://doi.org/10.1023/A:1009748529970

Molinaro, A. M., Simon, R., & Pfeiffer, R. M. (2005). Prediction error estimation: a comparison of resampling methods. Bioinformatics, 21(15), 3301–3307.
https://doi.org/10.1093/bioinformatics/bti499

Stevens, A., Nocita, M., Tóth, G., Montanarella, L., & van Wesemael, B. (2013). Prediction of Soil Organic Carbon at the European Scale by Visible and Near InfraRed Reflectance Spectroscopy. PLoS ONE, 8(6), e66409. https://doi.org/10.1371/journal.pone.0066409

Varma, S., & Simon, R. (2006). Bias in error estimation when using cross-validation for model selection. BMC Bioinformatics, 7, 91. https://doi.org/10.1186/1471-2105-7-91

Wetterlind, J., Stenberg, B., & Rossel, R. A. V. (2013). Soil Analysis Using Visible and Near Infrared Spectroscopy. In F. J. M. Maathuis (Ed.), Plant Mineral Nutrients (pp. 95–107). Totowa, NJ: Humana Press. https://doi.org/10.1007/978-1-62703-152-3_6

---

## Referee Report (RR1)

**General comments**

Most comments on technical and minor topical suggestions were addressed in the new manuscript.

5 Authors' reply: *"The PLSR components vary largely in dependence on the pre-processing method. The information on the number of selected components, therefore, did not result very informative. Furthermore, it distracts the reader from the main message. We, therefore, refrain from including it in this publication. We have adapted the results section concerning the uncertainty scenarios. We have also extended the introduction section to elaborate on the various sources of uncertainty.".*

I insist on giving finally selected tuning parameters for all the modeled scenarios. The reader requires such information to
10 judge the uncertainty in relation to model complexity within different error propagation scenarios. Hyperparameter optimization through Model tuning is a key aspect of spectral modeling, and all scientifically-sound publications in this field report these results. Parameter selection results can easily be included within a model assessment table. This does not distract the reader. This information will reveal also potential over-fitting effects due to replicate spectra in the inner tuning loop in presence of multiple spectral replicates.

15 The main concern from the last review round has been mentioned in the conclusion, but is still not resolved. Specifically, the group stratification for replicates of the same sample was not performed for the tuning procedure. The author's specifically mention this now in the conclusion. However, the author's refuse to report the finally selected PLSR parameters, but at the same time admit that suboptimal model parameters might be selected. Based on the results shown, no conclusive statement on this issue can be drawn.

20 In order to get this manuscript eligible for final publication, the issues arising from multiple replicates in tuning during cross-validation needs to be addressed. At least, the authors should report a majority consensus value of the number of PLSR components selected in the final model. This will at least indicate whether the model errors are biased by resampling artifacts.

**Specific comments**

*Abstract*

25 The statement in l. 24–26 about precise protocol and measurement protocol is out of scope and needs to be removed. It is not the main topic of this study. Such protocols and procedures are mostly well documented in soil spectroscopy literature and there are recommendations on this, consider for example Wetterlind et al., 2013.

*Introduction*

The introduction needs a major rewrite, there are many grammatical, topical, and stylistic errors. See the technical corrections
30 for some examples and suggestions.

*Material and Methods*

Authors' reply: *"Reference to soil treatment and scattering effects was made in the introduction. We refrain from referring to soil texture as we are at within-field scale and do not have a pronounced textural variability in our dataset. A reference to sample origin is included in the discussion section."*

Although the present study covers within-field variability, general conclusions regarding spectral error propagation need to consider soil texture as it affects scattering and averaging effects.

*Results and Discussion*

Some paragraphs contain statements that should be in the Material and Methods section, see e.g. lines 269–271 (see also technical corrections). Table 3 and the corresponding text needs to be moved to the Material and Methods section.

Many sections contain present tense where past tense is needed.

The section 3.1 on soil organic carbon reference values is too long, and needs some general revision. This section needs to be further summarized and presented in a more concise manner.

L. 283–284: "The plots for "A" and all samples show higher and lower SOC values than the archive data due to the fact that those data are obtained from compound samples for one plot.': It it not clear what the authors mean by compound plots..

Authors' reply: We are not aware of any study that actually quantified the effect of spectral pre-processing on model performance and, therefore, refrain from deleting it from our study.

This is simply not correct. There are many soil spectral studies addressing pre-processing with regard to model performance. Please consider e.g. Stevens et al., 2013 (see References at the end).

The discussion around the pre-processing is still way to long.

Table 4 on $R^2$ values is not informative. First, it only reports $R^2$, which is strictly not a measure of performance. Second, it is relative to the range of the measured property, which is not given. The only reference to this table is that the error conditional on the input data is different, and this information is missing in the table. Therefore, this table and discussion around it should be removed.

In general, the authors should stay focused on the key topics under investigation. Many sections are too long and therefore distract the reader. The Results and Discussion requires some more work to offer the audience a better flow.

*Conclusion*

The conclusion should fit on half a page. Focus on the key findings and topics that the study addressed.

**Technical corrections**

L. 12: "...the exact monitoring of..." -> "...precise monitoring of..."

L. 14 "...to enhance conventional SOC analysis and has often been used to predict SOC" -> "...to complement conventional SOC analysis."

L. 24–26: "We emphasize...and allow for a comparison between publications."

L. 35: "production of energy": Energy cannot be produced, rather use "production of energy crops"

L. 36: remove Stenberg et al., 2010: this does not fit the context.

L. 36–37: "quality of soil" needs to be described, too generic -> soil properties, soil type...

L. 38–40: "...SOC is also interesting when it comes to the global warning issue..." -> "SOC is also relevant for the global warning issue..."

L. 66: "However, the application of ....". -> Delete "However" because there is no reference sentence.

L. 67: "...standard lab analysis" -> "...standard laboratory analysis"

5  L. 68: "...on the other hand side" -> either "hand" or "side"

... (see general comments for abstract)

L. 100: Missing dot after "(Merbach and Schultz, 2013)"

L. 131: "Elementaranalysator" -> "elemental analyser"

L. 155–160: Give original publications for all pre-processing techniques; only give these and remove the other references.

10  L. 160–182: All references that are not original method publications for pre-processing need to be removed.

L. 276: "the SOC distribution of "A" and "B" samples differ" -> "the SOC distribution of "A" and "B" samples differed"

L. 307–308: "The model results are now compared based on their mean $RMSE_{MV}$ and their interquartile range": this should be in Material and Methods...

L. 308: "It is not surprising that the dataset...": style -> use "We expected that/It was expected that..." or similar

15  L. 312: "It seems that the within sample variation concerning soil spectra con somehow compendate the within sample variability concerning SOC within the model building process, although replicate measurements do not match" -> consistently use past tense.

L. 387: "cross validation" -> "cross-validation"

All figures need to be in vector graphics format or need a better resolution.

20  Figure 9: The text annotation for "RMSE" must be changed from "RMSE <number>" to "RMSE = <number>"

**References**

Stevens, A., Nocita, M., Tóth, G., Montanarella, L., van Wesemael, B. (2013). Prediction of Soil Organic Carbon at the European Scale by Visible and Near InfraRed Reflectance Spectroscopy. PLoS ONE, 8(6), e66409. https://doi.org/10.1371/journal.pone.0066409

Wetterlind, J., Stenberg, B., Rossel, R. A. V. (2013). Soil Analysis Using Visible and Near Infrared Spectroscopy. In F. J. M.
25  Maathuis (Ed.), Plant Mineral Nutrients (pp. 95–107). https://doi.org/10.1007/978-1-62703-152-3_6

**References**

REFERENCE 1
REFERENCE 2

---

## Author Response (AR2)

**General comments**

Most comments on technical and minor topical suggestions were addressed in the new manuscript.

Authors' reply: "The PLSR components vary largely in dependence on the pre-processing method. The information on the number of selected components, therefore, did not result very informative. Furthermore, it distracts the reader from the main message. We, therefore, refrain from including it in this publication. We have adapted the results section concerning the uncertainty scenarios. We have also extended the introduction section to elaborate on the various sources of uncertainty." I insist on giving finally selected tuning parameters for all the modeled scenarios. The reader requires such information to judge the uncertainty in relation to model complexity within different error propagation scenarios. Hyperparameter optimization through Model tuning is a key aspect of spectral modeling, and all scientifically-sound publications in this field report these results. Parameter selection results can easily be included within a model assessment table. This does not distract the reader. This information will reveal also potential over-fitting effects due to replicate spectra in the inner tuning loop in presence of multiple spectral replicates. The main concern from the last review round has been mentioned in the conclusion, but is still not resolved. Specifically, the group stratification for replicates of the same sample was not performed for the tuning procedure. The author's specifically mention this now in the conclusion. However, the author's refuse to report the finally selected PLSR parameters, but at the same time admit that suboptimal model parameters might be selected. Based on the results shown, no conclusive statement on this issue can be drawn. In order to get this manuscript eligible for final publication, the issues arising from multiple replicates in tuning during cross-validation needs to be addressed. At least, the authors should report a majority consensus value of the number of PLSR components selected in the final model. This will at least indicate whether the model errors are biased by resampling artifacts.

Reply: We decided to reply to these general comments collectively as they are all related to one another and refer to the applied nested cross-validation (CV) approach. There seems to be a misunderstanding. Model validation and tuning are both conducted with a group CV assigning replicate sample measurements and scans to the same fold. We adapted the corresponding section in the Materials and Methods and discussion section for better understandability. Please compare lines 248-283 and lines 405-408. "We agree that model complexity should always be kept in mind. However, overfitting was prevented by the applied nested group cross-validation (CV) approach. On the other hand, the mere inclusion of a table giving the number of components per model scenario would not suffice, as this would definitely require an extended discussion section, which is out of scope of this manuscript."

**Specific comments**

**Abstract**

The statement in l. 24–26 about precise protocol and measurement protocol is out of scope and needs to be removed. It isnot the main topic of this study. Such protocols and procedures are mostly well documented in soil spectroscopy literature and there are recommendations on this, consider for example Wetterlind et al., 2013.

Reply: Although there are protocols and procedures – we cite Pimstein et al. 2011 in lines 423-425 - there is still no agreement within the soil spectroscopy community on the applied protocol and procedure. As a consequence, the number of scans and replicate measurements per sample differ in each working group. As we have shown that the applied procedure has an impact on model performance, it is important to describe it in each study in order to allow for comparison between studies.We adapted the text section on Table 4 (lines 414-439) to that extent. As a consequence, the statement is in fact a major conclusion from this manuscript.

**Introduction**

The introduction needs a major rewrite, there are many grammatical, topical, and stylistic errors. See the technical corrections for some examples and suggestions.

Reply: Thank you, we have checked the introduction, thoroughly.

**Material and Methods**

Authors' reply: "Reference to soil treatment and scattering effects was made in the introduction. We refrain from referring to soil texture as we are at within-field scale and do not have a pronounced textural variability in our dataset. A reference to sample origin is included in the discussion section." Although the present study covers within-field variability, general conclusions regarding spectral error propagation need to consider soil texture as it affects scattering and averaging effects.

Reply: Soils with different properties may have a different amount of light scattering and might therefore require a differing amount of scans and replicate measurements per sample. However, this is beyond the scope of this paper. Based on the soil samples we measured we cannot make any statement in this regard.

**Results and Discussion**

Some paragraphs contain statements that should be in the Material and Methods section, see e.g. lines 269–271 (see also technical corrections). Table 3 and the corresponding text needs to be moved to the Material and Methods section.

Reply: We checked the Results and Discussion section, thoroughly. Table 3 and the corresponding text were moved to the Material and Methods section.

Many sections contain present tense where past tense is needed.

Reply: adapted accordingly

The section 3.1 on soil organic carbon reference values is too long, and needs some general revision. This section needs to be further summarized and presented in a more concise manner.

Reply: The text section was revised and summarised.

L. 283–284: "The plots for "A" and all samples show higher and lower SOC values than the archive data due to the fact that those data are obtained from compound samples for one plot.': It it not clear what the authors mean by compound plots..

Reply: The term "compound sample" was explained. The sentence was adapted to "The plots for "A" and all samples show higher and lower SOC values than the archive data due to the fact that those data were

obtained from compound samples , i.e. a number of distributed soil samples were taken per LTFE plot and mixed before they were subjected to soil laboratory analysis." (lines 325-326).

Authors' reply: We are not aware of any study that actually quantified the effect of spectral pre-processing on model performance and, therefore, refrain from deleting it from our study. This is simply not correct. There are many soil spectral studies addressing pre-processing with regard to model performance. Please consider e.g. Stevens et al., 2013 (see References at the end).

Reply: Yes, you are right. There are quite some publications that compare model performance in dependence on various pre-processing methods. We have actually cited some of them. The difference in model performance, due to the applied pre-processing, is usually not explicitly reported, though, but could be calculated. Still pre-processing is such an important aspect in VIS-NIR spectrometry that we find it important to report its impact on model performance in relation to the other aspects we investigated. Furthermore, the applied pre-processing changes the impact of the uncertainty propagation; in some cases the typical pattern is even reversed (Please compare lines 366-374).

The discussion around the pre-processing is still way to long.

Reply: The discussion of pre-processing only relates to its impact on model performance (now lines 375-383). We find this short paragraph of adequate length.

Table 4 on R2 values is not informative. First, it only reports R2, which is strictly not a measure of performance. Second, it is relative to the range of the measured property, which is not given. The only reference to this table is that the error conditional on the input data is different, and this information is missing in the table. Therefore, this table and discussion around it should be removed.

Reply: $R^2$ is still the most reported metric when comparing performance results between publications. We included the SOC range and further information and adapted the corresponding text section (lines 414-439) so that the reason for its inclusion is understandable. It relates to the information content of error values and the applied measurement protocol.

In general, the authors should stay focused on the key topics under investigation. Many sections are too long and therefore distract the reader. The Results and Discussion requires some more work to offer the audience a better flow.

Reply: Thank you. We have thoroughly revised the whole manuscript.

**Conclusion**

The conclusion should fit on half a page. Focus on the key findings and topics that the study addressed.

Reply: Adapted accordingly

**Technical corrections**

L. 12: "...the exact monitoring of..." -> "...precise monitoring of..."

Reply: changed accordingly

L. 14 "...to enhance conventional SOC analysis and has often been used to predict SOC" -> "...to complement conventional SOC analysis."

Reply: changed accordingly

L. 24–26: "We emphasize...and allow for a comparison between publications."

Reply: changed accordingly

L. 35: "production of energy": Energy cannot be produced, rather use "production of energy crops"

Reply: changed accordingly

L. 36: remove Stenberg et al., 2010: this does not fit the context.

Reply: adapted accordingly

L. 36–37: "quality of soil" needs to be described, too generic -> soil properties, soil type...

Reply: adapted accordingly

L. 38–40: "...SOC is also interesting when it comes to the global warning issue..." -> "SOC is also relevant for the global warning issue..."

Reply: changed accordingly

L. 66: "However, the application of ....". -> Delete "However" because there is no reference sentence.

Reply: However relates to the previous paragraph

L. 67: "...standard lab analysis" -> "...standard laboratory analysis"

Reply: changed accordingly

L. 68: "...on the other hand side" -> either "hand" or "side" ... (see general comments for abstract)

Reply: changed accordingly

L. 100: Missing dot after "(Merbach and Schultz, 2013)"

Reply: changed accordingly

L. 131: "Elementaranalysator" -> "elemental analyser"

Reply: changed accordingly

L. 155–160: Give original publications for all pre-processing techniques; only give these and remove the other references. L. 160–182: All references that are not original method publications for pre-processing need to be removed.

Reply: Adpated accordingly.

L. 276: "the SOC distribution of "A" and "B" samples differ" -> "the SOC distribution of "A" and "B" samples differed"

Reply: changed accordingly

L. 307–308: "The model results are now compared based on their mean $RMSE_{MV}$ and their interquartile range": this should be in Material and Methods...

Reply: adapted accordingly

L. 308: "It is not surprising that the dataset...": style -> use "We expected that/It was expected that..." or similar

Reply: changed accordingly

L. 312: "It seems that the within sample variation concerning soil spectra con somehow compendate the within sample variability concerning SOC within the model building process, although replicate measurements do not match" -> consistently use past tense.

Reply: Thank you.We have thoroughly revised the results and discussion section.

L. 387: "cross validation" -> "cross-validation"

Reply: changed accordingly

All figures need to be in vector graphics format or need a better resolution.

Reply: When embedding vector graphics in the applied software for manuscript writing, they get automatically rasterised. Vector files will be provided for final publication.

Figure 9: The text annotation for "RMSE" must be changed from "RMSE <number>" to "RMSE = <number>"

Reply: adapted accordingly

[revised manuscript text omitted]